# WeiPer: OOD Detection using Weight Perturbations of Class Projections

**Maximilian Granz** [*]
Institute for Computer Science
Free University of Berlin
Arnimallee 7 14195 Berlin
`maximilian.granz@fu-berlin.de`

**Manuel Heurich** [*]
Institute for Computer Science
Free University of Berlin
Arnimallee 7 14195 Berlin
`manuel.heurich@fu-berlin.de`

**Tim Landgraf**
Institute for Computer Science
Free University of Berlin
Arnimallee 7 14195 Berlin
`tim.landgraf@fu-berlin.de`

## Abstract

Recent advances in out-of-distribution (OOD) detection on image data show that pre-trained neural network classifiers can separate in-distribution (ID) from OOD data well, leveraging the class-discriminative ability of the model itself. Methods have been proposed that either use logit information directly or that process the model's penultimate layer activations. With "WeiPer", we introduce perturbations of the class projections in the final fully connected layer which creates a richer representation of the input. We show that this simple trick can improve the OOD detection performance of a variety of methods and additionally propose a distance-based method that leverages the properties of the augmented WeiPer space. We achieve state-of-the-art OOD detection results across multiple benchmarks of the OpenOOD framework, especially pronounced in difficult settings in which OOD samples are positioned close to the training set distribution. We support our findings with theoretical motivations and empirical observations, and run extensive ablations to provide insights into why WeiPer works. Our code is available at: `https://github.com/mgranz/weiper`.

## 1 Introduction

Out-of-Distribution (OOD) detection has emerged as a pivotal area of machine learning research. It addresses the challenge of recognizing input data that deviates significantly from the distribution seen during training. This capability is critical because machine learning models, particularly deep neural networks, are known to make overconfident and incorrect predictions on such unseen data Hendrycks & Gimpel (2016). The need for OOD detection is driven by practical considerations. In real-world applications, a model frequently encounters data that is not represented in its training set. For instance, in autonomous driving, a system trained in one geographic location might face drastically different road conditions in another. Without robust OOD detection, these models risk making unsafe decisions Amodei et al. (2016).

Over the last few years, the field has made significant steps towards setting up benchmarks and open baseline implementations. Thanks to the efforts of the OpenOOD team Zhang et al. (2023b); Yang

---

[*]Equal contribution.

38th Conference on Neural Information Processing Systems (NeurIPS 2024).

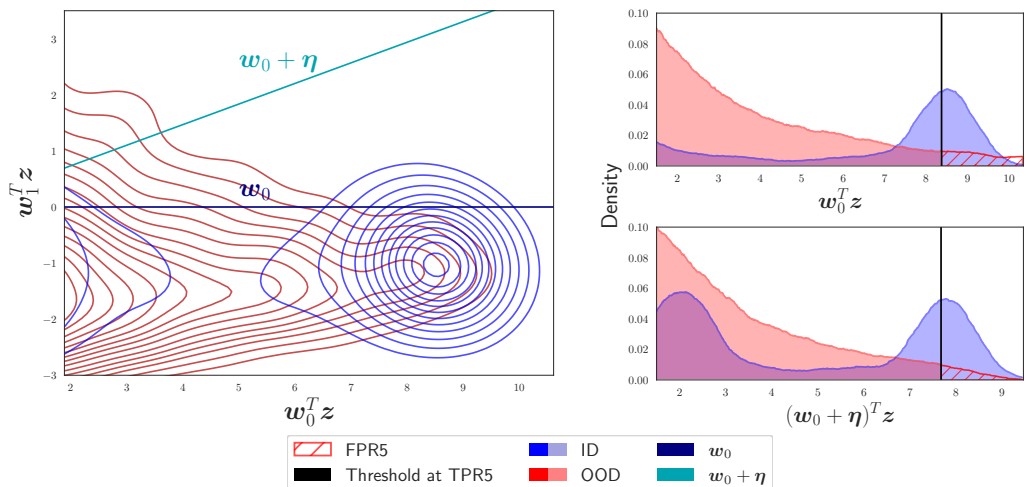

Figure 1: Why random perturbations? **Left**: We visualize densities of CIFAR10 (ID, blue) and CIFAR100 (OOD, red) as contour plots along the two logit dimensions spanned by $w_0$ and $w_1$, zoomed in on the positive cluster of class zero. The blue axis denotes the vector associated with that class, and one of its perturbations is depicted by the turquoise line. **Right**: When projecting the data onto both vectors, we obtain the densities shown in the *top* and *bottom* panel, respectively. The vertical blue lines mark the 5-percentile (highest 5%) of the true ID data (CIFAR10, blue). At this decision boundary, the classifier would produce false positives in the marked dashed red tail area. A single perturbation of the class-associated vector yields already a reduction of the false positive rate (FPR) from $1.34\%$ to $0.79\%$. Visually, we confirm that OOD data mostly resides close to 0, extending into the positive cluster in a particular conical shape, which is exploited by the cone of *WeiPer* vectors.

et al. (2022), we can evaluate new methods across CIFAR10, CIFAR100 and ImageNet, and compare them against a variety of methods, on the same network checkpoints. To this date, however, there is no single method outperforming the competition on all datasets Tajwar et al. (2021), which indicates a variety of ways in which OOD data differs from the training set. Here, we introduce *WeiPer*, a method that can be applied to any pretrained model, any training loss used, with no limitation on the data modality to separate ID and OOD datapoints. *WeiPer* creates a representation of the data by projecting the latent representation of the penultimate layer onto a cone of vectors around the class-projections of the final layer's weight matrix. This allows extracting additional structural information on the training distribution compared to using the class projections alone and specifically exploits the fact that the OOD data often extends into the cluster of positive samples of the respective class in a conical shape (see Figure 1). In addition to *WeiPer*, our KL-divergence-based method *WeiPer*+KLD represents a novel OOD detection score that is based on the following observation:

When ignoring the individual dimensions and examining the activation distribution across all dimensions, we observe that ID samples exhibit a similar "fingerprint" distribution. The more feature dimensions there are, the better our estimate of this source distribution becomes. We demonstrate that measuring the discrepancy between the per-sample distribution and the training set's mean distribution in the augmented *WeiPer* space leads to improved OOD detection accuracy. We evaluate *WeiPer* on OpenOOD using our proposed KL-divergence-based scoring function (**KLD**), MSP Hendrycks & Gimpel (2016), and ReAct Sun et al. (2021). Additionally, we conduct an ablation study to understand the influence of each component of *WeiPer* and analyze *WeiPer*'s performance. Our results confirm that the weight perturbations allow *WeiPer* to outperform the competition on two out of eight benchmarks, demonstrating consistently better performance on near OOD tasks. *WeiPer* represents a versatile, off-the-shelf method for state-of-the-art post-hoc OOD detection. However, the performance of *WeiPer* comes at a cost: The larger the *WeiPer* space, the more memory is required.

In summary, we present the following **contributions**:

- We discover that OOD detection can be improved by considering linear projections of the penultimate layer that correlate with the final output, i.e., the class representations. We construct these projections by perturbing the weights of the final layer.

- We uncover a fingerprint-like nature of the ID samples in both the penultimate space and our newly found perturbed space, proposing a novel post-hoc detection method that leverages this structure. The activation distributions of the penultimate space and our *WeiPer* space over the dimensions of each sample are similar for each ID input, yielding distributions in both spaces that we compare to the mean ID distribution using KL divergence.

- We evaluate our findings by testing the proposed methods and two other MSP-based methods on the perturbed class projections using the OpenOOD benchmark, achieving state-of-the-art performance on near OOD tasks.

## 2   Related work

**OOD detection.**   Generally, we can distinguish two types of OOD detection methods - one that requires retraining of the model, including novel loss variants, data augmentations, or even outlier exposure settings. Here, we focus on post-hoc methods that can be added with little effort to any existing pipeline. They can be applied to any pretrained model, irrespective of its architecture, loss objective or data modality. Post-hoc methods can be distinguished further in:

1) **Confidence-based** methods Guo et al. (2017); Hendrycks et al. (2022a,b); Liu et al. (2023a, 2020); Wang et al. (2022) process samples in the model's logit space, i.e. using the network directly to detect ID/OOD data points. A prominent example is the Maximum Softmax Probability (MSP) (Hendrycks & Gimpel, 2016) which simply uses the maximum logit as the main OOD decision metric. Some methods Ahn et al. (2023); Djurisic et al. (2022); Sun et al. (2021) additionally introduce transformations such as cutoffs of the features in the penultimate layer or masks on the weight matrix Sun & Li (2021) to allocate where ID data resides and combine these with confidence metrics. Several recent methods have employed f-divergences to improve OOD detection, focusing on enhancing the boundary definition between ID and OOD samples Darrin et al. (2022); Picot et al. (2022).

2) **Distance-based** methods Bendale & Boult (2015); Lee et al. (2018); Liu et al. (2023b); Ren et al. (2021); Sastry & Oore (2020); Sun et al. (2022); Zhang et al. (2023a) define distance measures between the training distribution and an input sample in latent space, i.e. primarily the penultimate layer of the network. Deep Nearest Neighbors Sun et al. (2022) uses the distance to the $k$-th closest neighbor in latent space, while MDS Lee et al. (2018) models the data as Gaussian and uses the Mahalonibis-Distance. Models of the data distribution can improve the OOD detection performance, e.g. using histograms to approximate the training density and then define a distance measure on them. A recent work Liu et al. (2023b) proposed creating a histogram-based distribution on the product of the penultimate activations and the gradient of a separate KL-loss and then defined a metric on these modified discrete densities.

Both approaches of 1) and 2) are not exclusive. NNGuide Park et al. (2023) combines both confidence and distance measures into a joint score, improving performance in case one of the scores fails.

**Random weight perturbations and projections.**   Weight perturbations, i.e. adding noise values to the weights of a network, have been used for a variety of applications: in sensitivity analyses Cheney et al. (2017); Xiang et al. (2019), for studying robustness against adversarial attacks Rakin et al. (2018); Wu et al. (2020), and as training regularization Khan et al. (2018); Wen et al. (2018). Random projections from the latent space of the neural network have been described in the context of generative modeling Bonneel et al. (2014); Jerome H. Friedman & Schroeder (1984); Kolouri et al. (2016); Liutkus et al. (2019); Nguyen et al. (2021); Paty & Cuturi (2019), e.g. to improve the Wasserstein distance calculation or for robustness. A previous work described random projections from the penultimate layer to detect out-of-distribution samples with a normalizing flow Kuan & Mueller (2022).

## 3 Method

### 3.1 Preliminaries

We consider a pretrained neural network classifier $f : \mathcal{X} \to \mathbb{R}^C$ that maps samples $x$ from an input space $\mathcal{X} \in \mathbb{R}^D$ to a logit vector $f(\boldsymbol{x}) \in \mathbb{R}^C$, by applying a linear projection $\boldsymbol{W}_{\text{fc}}$ to the feature representation in the penultimate layer

$$(z_1, ..., z_K)^T = \boldsymbol{z} = h(\boldsymbol{x}) = (h_1(\boldsymbol{x}), ..., h_K(\boldsymbol{x}))^T \tag{1}$$

such that $f(\boldsymbol{x}) = \boldsymbol{W}_{\text{fc}}^T \boldsymbol{z}$, with $D$, $K$ and $C$ representing the dimensionality of the input, the penultimate layer and the output layer, respectively. We define the rows of the final weight layer to be $\boldsymbol{w}_1, ..., \boldsymbol{w}_C$.

In the following it is useful to introduce $Z$ as random vector from which we draw our latent samples. We denote the densities of the latent activations of the training data with $p_{Z_{\text{train}}}$, of the test data with $p_{Z_{\text{test}}}$ and those of OOD samples with $p_{Z_{\text{ood}}}$ admitted by the random vectors $Z_{\text{train}}$, $Z_{\text{test}}$ and $Z_{\text{ood}}$, respectively. To ease notation, we will treat $Z$ and all its subsets, both as sets, e.g. $Z_{\text{train}}$ is the set of all training activations in the penultimate layer.

An OOD detector is a binary classifier $O$ that decides if samples are drawn from an ID or OOD distribution by usually only considering samples drawn from $p_{Z_{\text{test}}}$ or $p_{Z_{\text{ood}}}$. Commonly, this is achieved by thresholding a scalar score function $S$.

$$O(\boldsymbol{x}) = \begin{cases} \text{ID} & \text{if } S(\boldsymbol{x}) > \lambda \\ \text{OOD} & \text{otherwise} \end{cases} \tag{2}$$

For MSP, the score function is simply the maximum softmax probability

$$S(\boldsymbol{x}) = \text{MSP}(\boldsymbol{x}) = \max_{i=1,...,C} \frac{e^{f(\boldsymbol{x})_i}}{\sum_{j=1}^C e^{f(\boldsymbol{x})_j}} =: \text{MSP}(f(\boldsymbol{x})). \tag{3}$$

Note, that for clarity, we define MSP also as a function of the logits. Other methods propose metrics on the penultimate layer, e.g. by incorporating distance measures between a given latent activation $\boldsymbol{z}$ of a new sample and the distribution of activations $p_{Z_{\text{train}}}$ of the training set.

### 3.2 WeiPer: Weight perturbations

A neural network classifier maps the data distribution to the distribution of the logits $\boldsymbol{W}_{\text{fc}} Z$. The training objective of the network ensures an optimal separation of classes and lets the model learn to exploit features in $Z$ specific to the training distribution. OOD samples, hence, often yield lower logit scores. Confidence methods leverage this property, but could potentially be improved by capturing more of the underlying distribution of the penultimate layer. A confidence score measures properties of the logit distribution $\boldsymbol{W}_{\text{fc}} Z$. Is there additional information in the penultimate layer of the network, and if so, how can we utilize it?

Applying the weight matrix $\boldsymbol{W}_{\text{fc}}$ to the penultimate space can be understood as $C$ projections of $Z$ onto the row vectors $\boldsymbol{w}$. According the Cramer-Wold theorem (Cramér & Wold (1936)), we can reconstruct the source density $p_Z$ from all one-dimensional linear projections, and Cuesta-Albertos et al. (2007) has shown that a K-dimensional subset of projections suffices (for more details see Appendix A.1.1). The question remains which projections extract the most relevant information?

Drawing vectors $\boldsymbol{w} \in W = \mathcal{N}(0, I)$ from a standard normal and projecting onto them often results in similar densities for ID and OOD data, i.e. $\boldsymbol{w}^T Z_{\text{train}} \approx \boldsymbol{w}^T Z_{\text{ood}}$, deteriorating detection performance (see Table 1, RP). This aligns with Papyan et al. (2020), suggesting limited information in the penultimate layer compared to the logits. We hypothesize that the latent distribution shows relevant structure only along certain dimensions. We applied PCA to the latent activations $Z$ and inspected the resulting projections. This analysis supports the notion that the informative dimensions lie in the directions of the class projections $\boldsymbol{w}_1, ..., \boldsymbol{w}_C$ (see Appendix A.1.3). Hence, we construct projections that correlate with these vectors but at the same time deviate enough to obtain new information.

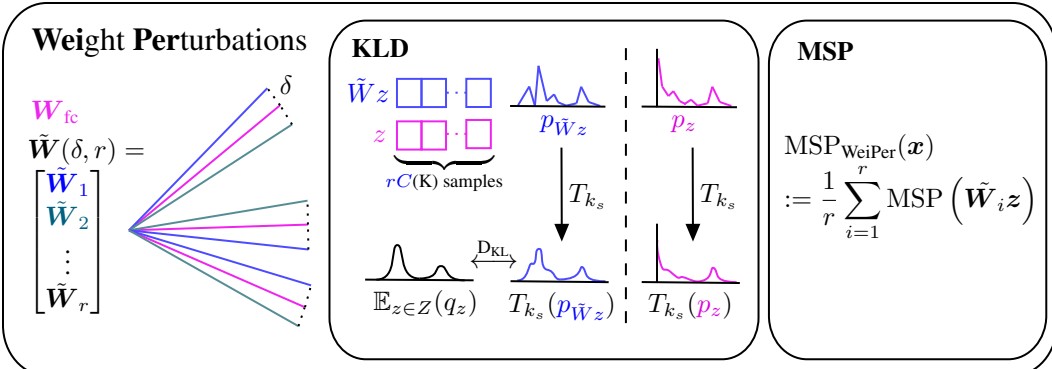

Figure 2: *WeiPer* perturbs the weight vectors of $W_{\text{fc}}$ by an angle controlled by $\delta$. For each weight, we construct $r$ perturbations resulting in $r$ weight matrices $\tilde{W}_1, ..., \tilde{W}_r$. **KLD**: For *WeiPer*+KLD, we treat $z_1, ..., z_k \sim p_z$ and $w_{1,1}^T z, ..., w_{r,C}^T z \sim p_{\tilde{W}z}$ as samples of the same distribution induced by $z$ and $\tilde{W}z$, respectively. We approximate the densities with histograms and smooth the result with uniform kernel $T_{k_s}$. Afterwards, we compare the densities $T_{k_s}(q_z)$ with the mean distribution over the training samples $\mathbb{E}_{z \in Z_{\text{train}}}(q_z)$ for $q_z = p_z$ and $q_z = p_{\tilde{W}z}$, respectively. **MSP**: For a score function $S$ on the logit space $\mathbb{R}^C$, we define the perturbed score $S_{\text{WeiPer}}$ as the mean over all the perturbed logit spaces $\tilde{W}z$. We choose $S = \text{MSP}$ and call the resulting detector $\text{MSP}_{\text{WeiPer}}$.

**Definition of WeiPer**  We define perturbations $\boldsymbol{\eta}$, drawn from a standard normal and add them to $W_{\text{fc}}$. To ensure that all perturbed vectors have the same angular deviation from the original weight vector, we normalize the perturbations to be the same length as their corresponding row vector and multiply them by a factor $\delta$:

$$\tilde{\boldsymbol{w}}_{i,j} = \boldsymbol{w}_j + \delta \frac{\boldsymbol{\eta}_i \|\boldsymbol{w}_j\|}{\|\boldsymbol{\eta}_i\|} =: \boldsymbol{w}_j + \tilde{\boldsymbol{\eta}}_i, \quad \boldsymbol{\eta}_i \sim \mathcal{N}(\boldsymbol{0}, \boldsymbol{I}_K) \tag{4}$$

for $i = 1, ..., r$, where $\delta$ represents the length ratio between $\boldsymbol{w}_j$ and the perturbation $\boldsymbol{\eta}_i$. For large $K = \dim(Z)$, $\boldsymbol{w}_j$ and $\tilde{\boldsymbol{\eta}}_i$ are almost orthogonal and thus $\delta$ actually adjusts the angle $\alpha \approx \arctan(\delta)$ of the perturbed vector bundle. We set $\delta$ to be constant across all $j = 1, ..., K$ and treat both $\delta$ and $r$ as hyperparameters. This proceedure is related to the Distributional Sliced Wasserstein distance Nguyen et al. (2021) as they sample projections from a distribution such that the mean angle between the projections is greater than $\arccos(C)$ for a constant $C$. The whole set of vectors we define is

$$W = \{\tilde{\boldsymbol{w}}_{1,1}, ..., \tilde{\boldsymbol{w}}_{1,C}, ..., \tilde{\boldsymbol{w}}_{r,C}\} \tag{5}$$

We can think of the resulting weight matrix $\tilde{W}$ as $r$ repetitions of the weight matrix $W_{\text{fc}}$ on which we add perturbation matrices $\tilde{H}_i$. The $j$-th row $\tilde{H}_{i,j}$ corresponds to a perturbation vector $\tilde{\boldsymbol{\eta}}_j$, normalized to match the respective row $\boldsymbol{w}_j$.

$$\tilde{W} := \begin{bmatrix} \tilde{W}_1 \\ \vdots \\ \tilde{W}_r \end{bmatrix} = \begin{bmatrix} W_{\text{fc}} + \tilde{H}_1 \\ \vdots \\ W_{\text{fc}} + \tilde{H}_r \end{bmatrix}, \tag{6}$$

Since $\tilde{W}_i Z = W_{\text{fc}} Z + H_i Z$, we call $\tilde{W} Z$ the perturbed logit space. Our weight perturbations method, we call WeiPer, essentially increases the output dimension of a model. Hence, it can be combined with many scoring functions. We demonstrate this with the two following postprocessors.

### 3.3  Baseline MSP scoring function

If the perturbations do not deviate too much from the class projections $\boldsymbol{w}_j$, i.e. the row vectors of the final layer, the class cluster will still be separated from the other classes in the new projections and we can apply MSP on the perturbed logit space. In fact, we find that class clusters on the perturbed projections can be better distinguished from the OOD cluster than on the original class projection

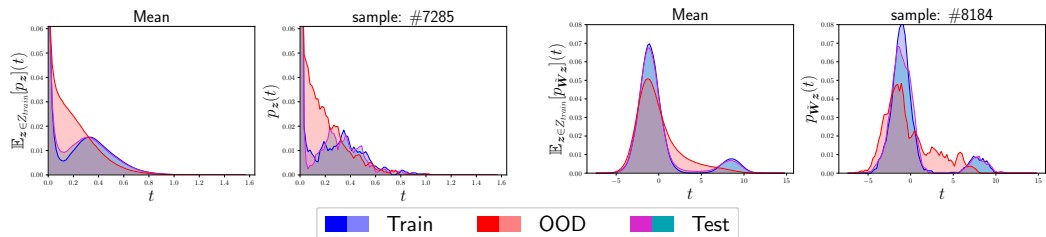

Figure 3: Histogram of all 512 activations in the penultimate layer (left pair) and the activations in *WeiPer* space (right pair) of a ResNet18 trained on CIFAR10. We perturb the weight matrix 100 times to produce a $10 \cdot 100 = 1000$-dimensional perturbed logit space. For each pair, the left panel shows the mean distribution over all samples (ID = CIFAR10, OOD = CIFAR100). The right panels show the distribution $p_z$ and $p_{\tilde{W}z}$, respectively, for two randomly chosen samples with smoothing applied ($s_1 = s_2 = 2$)

defined by $W_{\text{fc}}$ (see improvements of *WeiPer*+MSP($x$) over MSP in Table 2). Figure 1 illustrates a visual example. We calculate the MSP on the perturbed logit space as

$$\textit{WeiPer}+\text{MSP}(\boldsymbol{x}) := \text{MSP}_{\text{WeiPer}}(\boldsymbol{x}) := \frac{1}{r} \sum_{i=1}^{r} \text{MSP}(\tilde{\boldsymbol{W}}_i \boldsymbol{z}) \tag{7}$$

the mean over all the maximum softmax predictions of the perturbed logits. We analyze why $\text{MSP}_{\text{WeiPer}}$ could be capable of capturing more of the penultimate layer distribution than MSP in Appendix A.1.2.

## 3.4 Our KL divergence score function

Following our line of argument motivated by Theorem A.1, it seems natural to choose a density-based score function. When pooling all activations of the penultimate layer, an ID sample's activation distribution exhibits remarkable differences to that of an OOD sample. We observe the following properties:

- The majority of samples exhibit a bimodal distribution of their penultimate activations. An activation either belongs to the mode close to zero, or to the second mode (and rarely takes values in between).

- ID samples share a similar activation distribution. The mean activation distribution can serve as a prototype – see the upper left panel in Figure 3.

- The activation distribution is specific to the ID samples, i.e. the activation distribution of OOD samples differs from its distribution of ID samples and thus from the ID prototype.

Concluding on all three points, we make the *assumption* that all features $\boldsymbol{z} = h(\boldsymbol{x})$ of an ID input $x$ can be thought of as samples

$$z_1, ..., z_K \sim p_{\boldsymbol{z}}, \text{ where } (z_1, ..., z_K)^T = \boldsymbol{z}, \tag{8}$$

of the same underlying activation distribution $p_{\boldsymbol{z}}$. Furthermore, the density of $p_{\boldsymbol{z}}$ matches the mean distribution over all ID samples

$$p_{\boldsymbol{z}} \approx \mathbb{E}_{\boldsymbol{z}' \in Z_{\text{train}}}[p_{\boldsymbol{z}'}]. \tag{9}$$

We assume, the same is true for the logits. They naturally separate into a non-class cluster and a class cluster with the ratio $1 : C - 1$. Here, we could apply the same procedure, but especially for datasets with a small number of classes we would only get $C$ samples. This is where the cone of *WeiPer* vectors creates an advantage: They sit at a fixed angle to a class projection and thus preserve the class structure similarly across each projection onto one vector of the cone (e.g., like Figure 1 right - bottom panel). Analogous to Equation (9), we treat each projection

$$\boldsymbol{w}_{1,1}^T \boldsymbol{z}, ..., \boldsymbol{w}_{r,C}^T \boldsymbol{z} \sim p_{\tilde{\boldsymbol{W}}\boldsymbol{z}} \tag{10}$$

as a sample of the same underlying distribution and observe that

$$p_{\tilde{\boldsymbol{W}}\boldsymbol{z}} \approx \mathbb{E}_{\boldsymbol{z}' \in Z_{\text{train}}}[p_{\tilde{\boldsymbol{W}}\boldsymbol{z}'}]. \tag{11}$$

We demonstrate both behaviors in Figure 3.

In practice, we discretize $p_{\boldsymbol{z}}$ and $p_{\tilde{\boldsymbol{W}}\boldsymbol{z}}$ as histogram-based densities by splitting the value range into $n_{\text{bins}}$ bins (see Equation (21) in the Appendix). Compared to the mean distribution, $p_{\boldsymbol{z}}$ and $p_{\tilde{\boldsymbol{W}}\boldsymbol{z}}$ still have a sparse signal. We smoothen the densities with a function

$$T_{k_s}(p(t)) := \text{normalize}((p * k_s)(t) + \varepsilon) \tag{12}$$

by convolving $p$ with a uniform kernel $k_s$ of size $s$ and prevent densities from being zero by adding $\varepsilon > 0$ which we set to the fixed values $\varepsilon := 0.01$ for the penultimate layer and $\varepsilon := 0.025$ for the *WeiPer* space. Note, that tuning both epsilons might increase performance as we observed in early stages of our experiments, but will add two additional hyperparameters. We normalize the density to sum up to one again, here defined by $\text{normalize}$. Afterwards, we compare each of the densities with the KL divergence, respectively:

$$D_{\text{KL}}(\boldsymbol{x} \mid q_{\boldsymbol{z}}, k_s, \varepsilon) := \text{KL}\big(T_{k_s}(q_{\boldsymbol{z}}) \,\|\, \mathbb{E}_{\boldsymbol{z} \in Z_{\text{train}}}[q_{\boldsymbol{z}}]\big) + \text{KL}\big(\mathbb{E}_{\boldsymbol{z} \in Z_{\text{train}}}[q_{\boldsymbol{z}}] \,\|\, T_{k_s}(q_{\boldsymbol{z}})\big), \tag{13}$$

where $q_z$ is either $p_{\boldsymbol{z}}$ or $p_{\tilde{\boldsymbol{W}}\boldsymbol{z}}$. We discuss why our method does not suffer from the curse-of-dimensionality in contrast to other methods as investigated by Ghosal et al. (2023) in Appendix A.1.5

*WeiPer*+KLD combines the KL divergence on the penultimate space, the KL divergence and MSP on the perturbed logit space into one final score:

$$\textit{WeiPer}\text{+KLD}(\boldsymbol{x}) := D_{\text{KL}}(x \mid p_{\boldsymbol{z}}, s_1) + \lambda_1 D_{\text{KL}}(x \mid p_{\tilde{\boldsymbol{W}}\boldsymbol{z}}, s_2) - \lambda_2 \, \text{MSP}_{\text{WeiPer}}(\boldsymbol{x}) \tag{14}$$

The full list of hyperparameters is $r$ and $\delta$ for the *WeiPer* application and $n_{\text{bins}}, \lambda_1, \lambda_2, s_1, s_2$ for the KL divergence score function. Figure 2 provides a visual explanation and a quick overview of *WeiPer* and both its postprocessors.

## 4 Experiments

**Setup.** We evaluate *WeiPer* using the OpenOOD Zhang et al. (2023b) framework that includes three vision benchmarks: *CIFAR10* Krizhevsky (2009), *CIFAR100* Krizhevsky (2009), and *ImageNet* Deng et al. (2009). Each of them contains a respective ID dataset $\mathcal{D}_{\textbf{in}}$ and several OOD datasets, subdivided into *near* datasets $\mathcal{D}_{\textbf{near}}$ and *far* datasets $\mathcal{D}_{\textbf{far}}$ (see Table 1). The terms near and far indicate their similarity to $\mathcal{D}_{\textbf{in}}$ and, therefore, the difficulty of separating their samples.

OpenOOD also provides three model checkpoints trained on each CIFAR dataset whereas for ImageNet the methods are evaluated on a single official *torchvision* Marcel & Rodriguez (2010) checkpoint of ResNet50 He et al. (2016) and ViT-B/16 Dosovitskiy et al. (2020) respectively. We report our scores together with the results of Zhang et al. (2023b) in Table 2.

Due to resource constraints, we only evaluate our methods on the models trained with the standard preprocessor, that includes random cropping, horizontal flipping and normalizing, on the cross entropy objective. Additionally to the KL divergence score function and MSP, we evaluate *WeiPer* on ReAct. But instead of combining ReAct with the energy-based score function Liu et al. (2020) as in OpenOOD, we apply $\text{MSP}_{\text{WeiPer}}$ and call it *WeiPer*+ReAct. The hyperparameters of our methods were tuned by finding the best combination over a predefined and discrete range of values on the OpenOOD validation sets to assure a fair comparison to the competition (see Table 8).

For ImageNet, results are based on a subset of the training data, comprising 300,000 randomly selected, balanced samples (300 per class). For an analysis across different training set sizes, refer to Table 6.

**Metrics.** We evaluate the methods with the Area Under the Receiver Operating characteristic Curve, AUROC, Bradley (1997) metric as a threshold-independent score and the FPR95 as a quality metric. The FPR95 score reports the False Positive Rate at the True Positive Rate threshold 95%.

Table 1: The individual benchmark datasets.

| $\mathcal{D}_{\text{in}}$ | **CIFAR10** | **CIFAR100** | **ImageNet-1k** |
|---|---|---|---|
| $\mathcal{D}_{\text{out}}^{\text{near}}$ | CIFAR100, TinyImageNet | CIFAR10, TinyImageNet | ssb-hard, ninco |
| $\mathcal{D}_{\text{out}}^{\text{far}}$ | MNIST, SVHN, Texture, Places365 | MNIST, SVHN, Texture, Places365 | iNaturalist, Texture, OpenImage-O |

Table 2: OOD Detection results of top performing methods on the CIFAR10, CIFAR100 and ImageNet-1K benchmarks (For a comparison with every other evaluated method of OpenOOD and standard deviation over the CIFAR models, see Appendices A.5 and A.6). The top performing results for each benchmark are displayed in **bold** and we underline the second best result. Due to *WeiPer's* random nature, we report the median AUROC score over 10 different seeds. For an easy comparison, we portray the following ablations for CIFAR10 which are seperated by a line: The KLD results are the *WeiPer*+KLD results without MSP and RP is *WeiPer*+KLD with weight independent random projections drawn from a standard Gaussian. While WeiPer+KLD performs strongly especially on near datasets using ResNet backbones, its performance deteriorates with ViTs (see Section 4 for discussion).

| Method | $\mathcal{D}_{\textbf{near}}$ AUROC ↑ | FPR95 ↓ | $\mathcal{D}_{\textbf{far}}$ AUROC ↑ | FPR95 ↓ | $\mathcal{D}_{\textbf{near}}$ AUROC ↑ | FPR95 ↓ | $\mathcal{D}_{\textbf{far}}$ AUROC ↑ | FPR95 ↓ |
|---|---|---|---|---|---|---|---|---|
| | *Benchmark: CIFAR10 / Backbone: ResNet18* | | | | *Benchmark: CIFAR100 / Backbone: ResNet18* | | | |
| NAC | - | - | **94.60** | - | - | - | **86.98** | - |
| RMDS | 89.80 | 38.89 | 92.20 | 25.35 | 80.15 | 55.46 | 82.92 | 52.81 |
| ReAct | 87.11 | 63.56 | 90.42 | 44.90 | 80.77 | 56.39 | 80.39 | 54.20 |
| VIM | 88.68 | 44.84 | 93.48 | 25.05 | 74.98 | 62.63 | 81.70 | **50.74** |
| KNN | **90.64** | **34.01** | 92.96 | 24.27 | 80.18 | 61.22 | 82.40 | 53.65 |
| ASH | 75.27 | 86.78 | 78.49 | 79.03 | 78.20 | 65.71 | 80.58 | 59.20 |
| GEN | 88.20 | 53.67 | 91.35 | 34.73 | 81.31 | 54.42 | 79.68 | 56.71 |
| MSP | 88.03 | 48.17 | 90.73 | 31.72 | 80.27 | 54.80 | 77.76 | 58.70 |
| **WeiPer+MSP** | 89.00 | 40.71 | 91.42 | 28.87 | 81.32 | 54.49 | 79.95 | 57.00 |
| **WeiPer+ReAct** | 88.83 | 42.84 | 91.23 | 29.50 | 81.20 | 55.03 | 80.31 | 55.61 |
| **WeiPer+KLD** | 90.54 | 34.06 | 93.12 | **23.72** | **81.37** | **54.34** | 79.01 | 57.96 |
| KLD | 90.53 | 34.12 | 93.15 | 23.58 | 76.68 | 66.41 | 68.95 | 71.70 |
| RP | 69.62 | 87.72 | 75.83 | 75.66 | 70.68 | 73.98 | 67.23 | 77.25 |

| Method | $\mathcal{D}_{\textbf{near}}$ AUROC ↑ | FPR95 ↓ | $\mathcal{D}_{\textbf{far}}$ AUROC ↑ | FPR95 ↓ | $\mathcal{D}_{\textbf{near}}$ AUROC ↑ | FPR95 ↓ | $\mathcal{D}_{\textbf{far}}$ AUROC ↑ | FPR95 ↓ |
|---|---|---|---|---|---|---|---|---|
| | *Benchmark: ImageNet-1K / Backbone: ResNet50* | | | | *Benchmark: ImageNet-1K / Backbone: ViT-B/16* | | | |
| NAC | - | - | 95.29 | - | - | - | **93.16** | - |
| RMDS | 76.99 | 65.04 | 86.38 | 40.91 | **80.09** | 65.36 | 92.60 | **28.76** |
| React | 77.38 | 66.69 | 93.67 | 26.31 | 69.26 | 84.49 | 85.69 | 53.93 |
| VIM | 72.08 | 71.35 | 92.68 | 24.67 | 77.03 | 73.73 | 92.84 | 29.18 |
| KNN | 71.10 | 70.87 | 90.18 | 34.13 | 74.11 | 70.47 | 90.81 | 31.93 |
| ASH | 78.17 | 63.32 | **95.74** | **19.49** | 53.21 | 94.43 | 51.56 | 96.77 |
| GEN | 76.85 | 65.32 | 89.76 | 35.61 | 76.30 | 70.78 | 91.35 | 32.23 |
| MSP | 76.02 | 65.68 | 85.23 | 51.45 | 73.52 | 81.85 | 86.04 | 51.69 |
| **WeiPer+MSP** | 77.68 | 63.84 | 89.33 | 41.56 | 74.82 | 74.97 | 89.15 | 43.49 |
| **WeiPer+ReAct** | 76.85 | 66.87 | 93.09 | 29.83 | 74.79 | 74.08 | 89.45 | 41.22 |
| **WeiPer+KLD** | **80.05** | **61.39** | 95.54 | 22.08 | 75.00 | 73.02 | 90.32 | 38.16 |

**Results.** Table 2 reports the performance of *WeiPer* in comparison to the state-of-the-art OOD detectors on each benchmark. We compare our approach based on the $\mathcal{D}_{\textbf{near}}$ and $\mathcal{D}_{\textbf{far}}$ detection performances and report the mean over all datasets in each category. Table 3 portrays the mean relative performance on $\mathcal{D}_{\textbf{near}}$ and $\mathcal{D}_{\textbf{far}}$ of every postprocessor. The score is calculated as follows:

$$S_{\text{rel}}(P) := \frac{1}{3}\big(A_{\text{CIFAR10}}(P) + A_{\text{CIFAR100}}(P) + \frac{1}{2}\big(A_{\text{ImageNet(ResNet50)}}(P) + A_{\text{ImageNet(ViT))}}(P)\big)\big) \quad (15)$$

where

$$A_{\mathcal{D}}(P) := \frac{\text{AUROC}_{\mathcal{D}_{\text{near/far}}}(P)}{\max_{P \in \mathcal{P}} \text{AUROC}_{\mathcal{D}_{\text{near/far}}}(P)} \quad (16)$$

It is designed such that each result on each dataset $\mathcal{D}$ is equally weighted and scoring 1.0 means that the postprocessor $P$ is top performing across all datasets.

*WeiPer*+KLD achieves three out of eight top AUROC scores and the best performance on all near benchmarks, establishing a new state of the art performance by a significant margin (see Table 3).

Especially for the most challenging benchmark, separating $\mathcal{D}_{\textbf{near}}$ on ImageNet with a ResNet50, we outperform our strongest competitor, ASH Djurisic et al. (2022), by 1.88% AUROC (we even achieve an AUROC score of 80.29 when using a 1M training samples instead of 300k, see Table 6). Additionally, *WeiPer*+KLD performs well on many far benchmarks, being the best method for ResNet50 on ImageNet, reaching into the top three positions on CIFAR10 far and into the top three on the CIFAR100 far benchmark. With its relative performance in Table 3, *WeiPer*+KLD reaches 3rd place overall in the far benchmark.

Only on Vit-B/16 trained on ImageNet, *WeiPer*+KLD shows a significant performance dent, especially on the far benchmark. ViT-B/16 uses a comparably narrow penultimate layer having fewer features

Table 3: Mean relative scores of all the postprocessors (post-hoc methods), see Equation (15).

| $\mathcal{D}_{\text{near}}$ | | | | $\mathcal{D}_{\text{far}}$ | | | |
|---|---|---|---|---|---|---|---|
| **Postprocessor** | $S_{\text{rel}}$ | **Postprocessor** | $S_{\text{rel}}$ | **Postprocessor** | $S_{\text{rel}}$ | **Postprocessor** | $S_{\text{rel}}$ |
| **WeiPer+KLD** | 0.988 | OpenMax | 0.943 | NAC | 0.999 | MLS | 0.932 |
| RMDS | 0.984 | VIM | 0.943 | VIM | 0.970 | TempScale | 0.929 |
| **WeiPer+MSP** | 0.977 | EBO | 0.940 | KNN | 0.963 | EBO | 0.924 |
| GEN | 0.975 | SHE | 0.934 | **WeiPer+KLD** | 0.959 | MSP | 0.920 |
| **WeiPer+ReAct** | 0.974 | KLM | 0.918 | RMDS | 0.959 | SHE | 0.919 |
| TempScale | 0.967 | DICE | 0.901 | **WeiPer+ReAct** | 0.951 | DICE | 0.909 |
| KNN | 0.963 | ASH | 0.870 | GEN | 0.947 | KLM | 0.893 |
| MSP | 0.963 | MDS | 0.829 | **WeiPer+MSP** | 0.944 | MDS | 0.877 |
| ReAct | 0.955 | GradNorm | 0.722 | ReAct | 0.943 | ASH | 0.844 |
| MLS | 0.954 | NAC | - | OpenMax | 0.935 | GradNorm | 0.700 |

than classes and therefore compresses the class clusters. Some dimensions may thus compress two classes while others represent a feature specific to only one class. This introduces more noise into $p_{\mathbf{z}}$ which could impair the detection performance. Future experiments will reveal whether *WeiPer* benefits from higher dimensionalities of the latent space.

**WeiPer on existing methods.**  Additionally, *WeiPer* enhances the MSP performance by 1-4.1% AUROC across all benchmarks and *WeiPer*+ReAct consistently outperforms ReAct with an energy-based score, although in their evaluation, this variant was better than ReAct+MSP (see Table 3).

**Ablation study.**  We determine the effect of each hyperparameter in Figure 4 by freezing single hyperparameters and optimizing only the one in question. As expected, increasing the number of random perturbations $r$ leads to a better median performance, while the standard deviation decreases for larger $r$. Note, that it is possible to have better performance for lower $r$ by rerolling the weights a few times and choosing the best performing ones. All methods show a significant performance boost compared to using no perturbations $\delta = 0$ and seem to be best at $\delta = 2$ for *WeiPer*+KLD, which corresponds to an angle of $\alpha \approx 63°$ and $\delta = 4$ ($\alpha \approx 76°$) for MSP and ReAct.

On CIFAR10, *WeiPer*+KLD only improves marginally by applying $\text{MSP}_{\text{WeiPer}}$, which is not the case for the other benchmarks (see Table 7), where $\lambda_2 > 0$. Furthermore, we study the performance of random projections that are independent from the weights $W_{\text{fc}}$. We show that using only random projections (RP, see Table 2) without adding $\text{MSP}_{\text{WeiPer}}$, we are hardly able to detect any OOD samples. This supports the claim that utilizing the class directions is necessary. The supplementary material presents all the other KLD-specific hyperparameters and we also investigate their influences to the performance in Figure 6. We outline the selected parameters for each benchmark in Table 7.

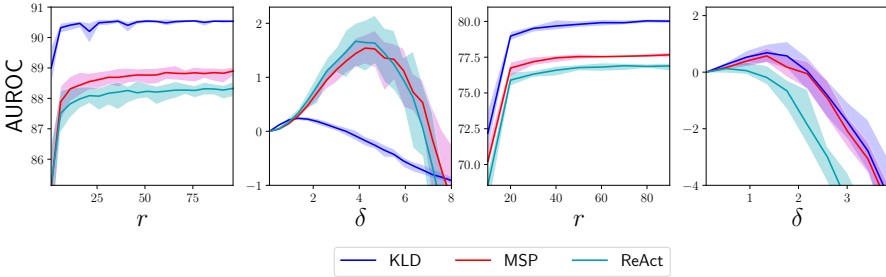

Figure 4: We investigate the effect of *WeiPer* hyperparameters $r$ and $\delta$ on the performance of the three postprocessors. The left pair shows results on CIFAR10, the right pair corresponds to ImageNet (using ResNet18 for both). Models were tested using their respective near OOD datasets. The panels corresponding to $\delta$ depict AUROC performance minus the initial AUROC performance at $\delta = 0$. The graphs show the mean over 25 runs and the shaded area around them represents the value range (min to max) over those runs. All other parameters of the methods were fixed to the optimal setting.

## 5 Limitations

*WeiPer*+KLD has more hyperparameters than other competitors: 6 in total. As discussed in the previous section, $r$ can be seen as a memory / performance trade-off (see Figure 4). In the supplementary material (see Figure 6) we investigate the other parameters and find that they all have only one local maximum in the range we were searching and should thus be easy to optimize. We tried to choose the same smoothing size $s_1 = s_2$ for both densities, but the ablations show that both are optimal at different sizes. While $\mathrm{MSP}_{\mathrm{WeiPer}}$ is not really used for CIFAR10 ($\lambda_2 \approx 0$) it is beneficial for CIFAR100 and ImageNet. As WeiPer blows up the dimension we also conduct a memory and time comparison to other methods in Table 4 and Table 5. We demonstrate that with a combination of a confidence and a distance based metric it is possible to achieve competitive near results across the board where all other methods seem to deteriorate in at least one benchmark.

## 6 Conclusion

We show that multiple random perturbations of the class projections in the final layer of the network can provide additional information that we can exploit for detecting out-of-distribution samples. *WeiPer* creates a representation of the data by projecting the latent activation of a sample onto vector bundles around the class-specific weight vectors of the final layer. We then employ a new approach to construct a score allowing the subsequent separation of ID and OOD data. It relies on the fingerprint-like nature of features of the penultimate and the *WeiPer*-representations by assuming they were sampled by the same underlying distribution. In a thorough evaluation, we first show that *WeiPer* enhances MSP and ReAct+MSP performance significantly and show that *WeiPer*+KLD achieves top scores in most benchmarks, representing the new state-of-the-art solution in post-hoc OOD methods on near benchmarks.

## 7 Acknowledgements

We appreciate the reviewers' comments, which greatly helped enhance this manuscript and inspired us to conduct additional key experiments. Maximilian Granz was supported by the Elsa-Neumann-Scholarship from the state of Berlin, which provided essential funding for the initial stages of this research. We also thank Leon Sixt and Manolis Panagiotou for their feedback throughout the project.

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

# A Appendix

## A.1 WeiPer: Theoretical details

### A.1.1 Weight perturbations

**Theorem A.1** (Cuesta-Albertos et al. (2007)). *Let $X$ and $Y$ be two $\mathbb{R}^K$-valued random vectors. Suppose the absolute moments $m_k := \mathbb{E}(\|X\|^k)$ are finite and $\sum_{k=1}^{\infty}(m_k)^{-1/k} = \infty$. If the set $W_{XY} = \{\boldsymbol{w} \in \mathbb{R}^K : \boldsymbol{w}^T X \stackrel{d}{=} \boldsymbol{w}^T Y\}$ has positive Lebesgue measure, then $X \stackrel{d}{=} Y$.*

We provide a simple proof for the case that $W_{XY} = \mathbb{R}^K$. For the complete proof, we refer to Cuesta-Albertos et al. (2007).

*Proof.* The characteristic function of a random vector $X$ is defined as

$$\phi_X(\boldsymbol{w}) := \int e^{i\boldsymbol{w}^T \boldsymbol{x}} dX \tag{17}$$

By the uniqueness theorem, every random vector $X$ has a unique characteristic funtion $\phi_X$. If the assumption in the theorem holds, then for all realizations $\boldsymbol{x} = X(\omega)$ and $\boldsymbol{y} = Y(\omega)$, we have

$$\boldsymbol{w}^T \boldsymbol{x} = \boldsymbol{w}^T \boldsymbol{y} \tag{18}$$

and thus $\phi_X = \phi_Y$. Therefore we have $X = Y$ by the uniqueness. $\square$

Note, that is enough cover all the directions in $\mathbb{R}^K$ instead of covering the whole space with the set of projections $W$. Since if $t\boldsymbol{w} \notin W$ for $t \in \mathbb{R}$, but $w \in W \cap W_{XY}$ then $t\boldsymbol{w} \in W_{XY}$. For $\delta > 0$ our set of perturbed class projections indeed covers the all directions if $r \to \infty$.

To generally apply the theorem, $Z$ must be defined on a bounded set with finite measure (Hausdorff moment problem), which is true for virtually all practical problems. More importantly, $W$ needs to be a $K$-dimensional subset of $\mathbb{R}^K$. Note that the theorem also applies for a set of weight matrices

$$\{\boldsymbol{W} \in \mathbb{R}^{K \times K} : \boldsymbol{W} X \stackrel{d}{=} \boldsymbol{W} Y\} \tag{19}$$

when their row vectors form a $K$ dimensional set as their marginal distributions $\boldsymbol{w}_i^T X \stackrel{d}{=} \boldsymbol{w}_i^T Y$ would be equal for $i = 1, ..., K$. We are using this theorem solely as motivation since it is not possible to draw direct implications. However, with a score function $S$ we are measuring properties of the logit distribution of the training data $\boldsymbol{W}_{\text{fc}} Z_{\text{train}} = (\boldsymbol{w}_1^T Z_{\text{train}}, ..., \boldsymbol{w}_C^T Z_{\text{train}})$ and check if they match the properties of some unknown logit distribution $\boldsymbol{W}_{\text{fc}} Z$ that might be test data or OOD data. In the ideal case the logits match in distribution

$$\boldsymbol{W}_{\text{fc}} Z_{\text{train}} \stackrel{d}{=} \boldsymbol{W}_{\text{fc}} Z \text{ if and only if } S \text{ is maximized,} \tag{20}$$

e.g. if $D$ is a distance and $S = -D$, $S(\boldsymbol{W}_{\text{fc}} Z_{\text{test}}) = 0$ and $S(\boldsymbol{W}_{\text{fc}} Z_{\text{ood}}) < 0$. Now the theorem says that if we chose a $K$-dimensional set $W$ of projections and we had a score function $S_{\text{WeiPer}}$ that fulfills the property of Equation (20) on the infinite dimensional space spanned by the projections of $W$, not just the distributions on the projection would be equal when $S_{\text{WeiPer}}$ is minimized but also the penultimate distributions $Z_{\text{train}} \stackrel{d}{=} Z_{\text{test}}$.

### A.1.2 MSP on the perturbed logit space

Continuing from the previous section, $S_{\text{WeiPer}} = \text{MSP}_{\text{WeiPer}}$ is a score function on the infinite dimensional space spanned by $W$ for $r \to \infty$, so ideally $\text{MSP}_{\text{WeiPer}}(Z_{\text{test}}) = 0$ then not only the logit distributions match $\boldsymbol{W}_{\text{fc}} Z_{\text{train}} \stackrel{d}{=} \boldsymbol{W}_{\text{fc}} Z_{\text{test}}$, but also their penultimate distributions $Z_{\text{train}} \stackrel{d}{=} Z_{\text{test}}$ which would make $\text{MSP}_{\text{WeiPer}}$ a stronger metric than MSP.

### A.1.3 PCA on the penultimate space

We draw the following conclusions from Figure 5: The OOD and ID distributions differ much stronger along the first $C$ principal components, and they are more similar for the other components. This indicates, most of the signal may lie in the $C$-dimensional subspace.

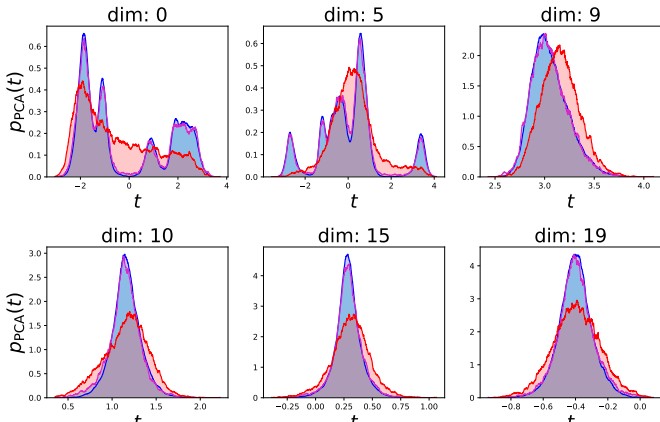

Figure 5: We applied PCA to $Z_{\text{Train}}$ of CIFAR10 and projected $Z_{\text{train}}$ (blue), $Z_{\text{test}}$ (purple) and $Z_{\text{ood}}$ (red, CIFAR100) to the first 20 principal components. We observe density spikes in the first 10 dimensions, likely corresponding to the class clusters. The dimensions 10-19 exhibit less structure as their densities appear to be Gaussian. Along these directions the ID and OOD data are more similar compared to the first ten principal components.

We argue that instead of taking random projections, we can utilize the class projections. Since we have a trained classifier at hand, it is likely that the informative dimensions are: $\text{span}(\boldsymbol{w}_1, ..., \boldsymbol{w}_C)$, the span of the row vectors of $\boldsymbol{W}_{\text{fc}}$. Hence, a better choice than Gaussian vectors for the set of projections $W$ are vectors $\boldsymbol{w}$ that correlate with these basis vectors in $\boldsymbol{W}_{\text{fc}}$ but at the same time deviate enough such that we obtain new information from projections onto them.

### A.1.4  KL divergence: Density definition

We gave a brief description of our construction of the densities in the main paper. The formal definition is:

$$p_{\boldsymbol{z}}(t) := \frac{1}{Kl_b} \sum_{i=1}^{K} [z_i \in b(t)]. \tag{21}$$

Here $l_b$ is the bin length, $b(t)$ is the bin range in which $t$ falls, and $[.]$ is the Iverson bracket which evaluates to one if true or zero if the statement is false. Note that we are dividing by $l_b$ such that the density integrates to one. This is the usual definition for descretizing a density into equal sized bins.

### A.1.5  Curse of dimensionality

In contrast to other distance-based methods, our KL divergence score does not suffer from the curse-of-dimensionality, which deteriorates the performance of methods like KNN Sun et al. (2022) as investigated by Ghosal et al. (2023). We disregard the dimension and only consider the activations in the penultimate space or in the perturbed logit space. In our case, more dimensions means more samples to approximate the activation distribution $p_{\boldsymbol{z}}$. We believe that our method thrives when applied to networks with higher dimensional penultimate space, but this still needs to be evaluated in future experiments. However, in the perturbed logit space, we can control the size of the space with $r$ (see Figure 4). Our ablation results show that increasing $r$ and thus blowing up the dimensionality only increases performance.

## A.2 Memory and Time Analysis

Table 4: Time taken in seconds to *setup* the method or for *inference*. More precisely, we measure the time of `postprocessor.setup()` and `postprocessor.inference()` in OpenOODs `evaluator.py` . We take the mean over 20 iterations and denote the standard deviation after the ± sign. We mark the maximal time in bold and underline the second longest time taken. We compare WeiPer+KLD ($r = 100$ as in the paper) to its closest competitors and show that it is competitive with other methods in terms of computation time. While WeiPer+KLD is on the higher end of the spectrum in terms of computation time, it remains comparable to other methods. It's worth noting that computation time can be adjusted by trading off performance with lower $r$ values.

| Time | Weiper+KLD | NAC | KNN | RMDS | VIM |
|------|-----------|-----|-----|------|-----|
| | | | *Setup* | | |
| CIFAR10 | 21.2 ± 0.09 | **24.4 ± 0.35** | 10.5 ± 0.03 | 11.2 ± 0.04 | 11.1 ± 0.05 |
| CIFAR100 | **25.2 ± 0.06** | 24.3 ± 0.24 | 10.5 ± 0.04 | 11.3 ± 0.03 | 11.1 ± 0.05 |
| ImageNet | 1975.1 ± 7.4 | **5676.9 ± 34.6** | 1599.1 ± 6.5 | 1631.7 ± 1.7 | 1636.1 ± 1.9 |
| | | | *Inference* | | |
| CIFAR10 | 63.2 ± 0.88 | 41.8 ± 0.24 | **71.2 ± 0.62** | 47.9 ± 0.2 | 53.1 ± 0.44 |
| CIFAR100 | **104.6 ± 1.37** | 41.4 ± 0.18 | 70.9 ± 0.98 | 97.2 ± 1.05 | 53.4 ± 0.7 |
| ImageNet | 882.0 ± 4.5 | 223.7 ± 0.9 | **14507.0 ± 86.6** | 1167.7 ± 8.1 | 273.3 ± 1.1 |

Table 5: Memory consumption in MiB to *setup* the method or for *inference*. More precisely, we compare the memory of `postprocessor.setup()` and `postprocessor.inference()` in OpenOODs `evaluator.py` before and after its execution with `psutil` . We take the mean over 20 iterations of the data between the 20% and 80% quantile to diminish the effect of outliers (e.g., caused by interfering processes) and denote the standard deviation after the ± sign. We mark the maximal memory in bold and underline the second highest demand. WeiPer+KLDs ($r = 100$) memory consumption for its setup is among the lower demanding methods while it has a comparably high demand for inference. Note that for optimal results we choose $r = 100$, but smaller values of $r$ also provide competitive results (see Figure 4). Thus memory can be traded against performance where resources are constraint.

| Memory | Weiper+KLD | NAC | KNN | RMDS | VIM |
|--------|-----------|-----|-----|------|-----|
| | | | *Setup* | | |
| CIFAR10 | 2602.6 ± 1 | 2557.3 ± 6 | 2736.6 ± 10 | **2806.0 ± 31** | 2725.1 ± 13 |
| CIFAR100 | 2717 ± 10.2 | 2556.7 ± 4 | 2732.8 ± 4 | **2763.5 ± 27** | 2726.3 ± 14 |
| ImageNet | 1464.1 ± 3 | 1871.6 ± 28 | **31744.9 ± 18** | 14965.1 ± 72 | 13025.0 ± 6 |
| | | | *Inference* | | |
| CIFAR10 | 5.7 ± 0.3 | 7.6 ± 1.9 | **12.0 ± 1.0** | 5.4 ± 2.7 | 4.9 ± 1.7 |
| CIFAR100 | **18.2 ± 8.9** | 5.6 ± 2.5 | 11.0 ± 1.3 | 4.3 ± 0.8 | 3.1 ± 0.1 |
| ImageNet | **121.4 ± 0.4** | 20.7 ± 0.1 | 19.0 ± 4.4 | 13.1 ± 0.1 | 6.6 ± 0.2 |

Table 6: AUROC results on ImageNet with ResNet50 on the near and far benchmark with different training set sizes. Each split is a random sample of the data set with each class appearing exactly as often as each other class. We chose the optimal set of hyperparameters on ImageNet, but reduced the number of repeats $r$ to 50 instead of 100.

| #Samples | 1k | 5k | 10k | 50k | 100k | 500k | 1M |
|----------|-----|-----|-----|-----|------|------|-----|
| $\mathcal{D}_{\text{near}}$ | 69.46 | 72.57 | 74.46 | 77.61 | 77.70 | 79.65 | 80.29 |
| $\mathcal{D}_{\text{far}}$ | 85.76 | 89.79 | 91.90 | 94.53 | 94.47 | 95.51 | 95.56 |

## A.3 Hyperparameters

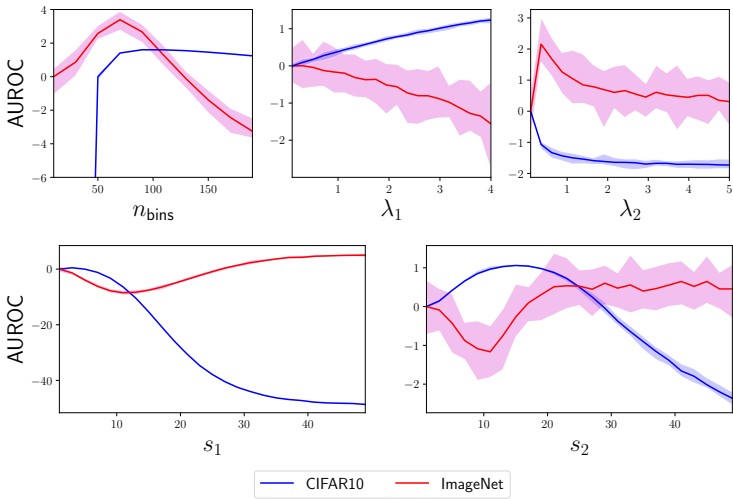

Figure 6: KLD specific hyperparamters: We fixed the optimal hyperparameters and varied the one parameter in question by conducting 10 runs over the same fixed parameter setting on CIFAR10 and ImageNet as ID against their near OOD datasets. We report the mean and the minimum to maximum range (transparent). We set $r = 5$ instead of $r = 100$ for ImageNet to save resources. Thus the noise on the results is stronger for the ImageNet ablations. All of the parameters except the kernel sizes only have a single local maximum which indicates that they should be easy to optimize. The most important parameters seem to be the kernel sizes $s_1$ and $s_2$ that we use for smoothing followed by $n_{\text{bins}}$. Note that $s_1$ and $s_2$ have a different optimum, which means it is not possible to simply choose $s_1 = s_2$ and reduce the count of hyperparameters. $\lambda_1 = 0$ is the score function without the KLD specific *WeiPer* application. $\lambda_2$ is the application of $\text{MSP}_{\text{WeiPer}}$ which is not beneficial for CIFAR10, but shows to be effective on ImageNet.

Table 7: The hyperparameter sets for each experiment. The number of repeats $r$ was predefined since we found increasing it always boosts the performance at the cost of time and memory consumption.

| Hyperparameter | CIFAR10 (ResNet18) | CIFAR100 (ResNet18) | ImageNet-1K (ResNet50) | ImageNet-1K (ViT-B/16) |
|---|---|---|---|---|
| $r$ | 100 | 100 | 100 | 100 |
| $\lambda$ | 1.8 | 2.4 | 2.4 | 2.0 |
| $n_{\text{bins}}$ | 100 | 100 | 100 | 80 |
| $\lambda_1$ | 2.5 | 0.1 | 2.5 | 2.5 |
| $\lambda_2$ | 0.1 | 1 | 0.25 | 0.1 |
| $s_1$ | 4 | 4 | 40 | 40 |
| $s_2$ | 15 | 40 | 15 | 15 |

Table 8: Set of values for the hyperparameter search.

| Hyperparameter | Values |
|:---:|:---|
| $r$ | 100 |
| $\lambda$ | [1.8, 2.0, 2.2, 2.4] |
| $n_{\text{bins}}$ | [60, 80, 100] |
| $\lambda_1$ | [0.1, 1, 2.5, 4] |
| $\lambda_2$ | [0.1, 0.25, 1, 2.5, 5] |
| $s_1$ | [4, 8, 12, 20, 40] |
| $s_2$ | [15, 25, 40] |

### A.4 Penultimate layer distribution

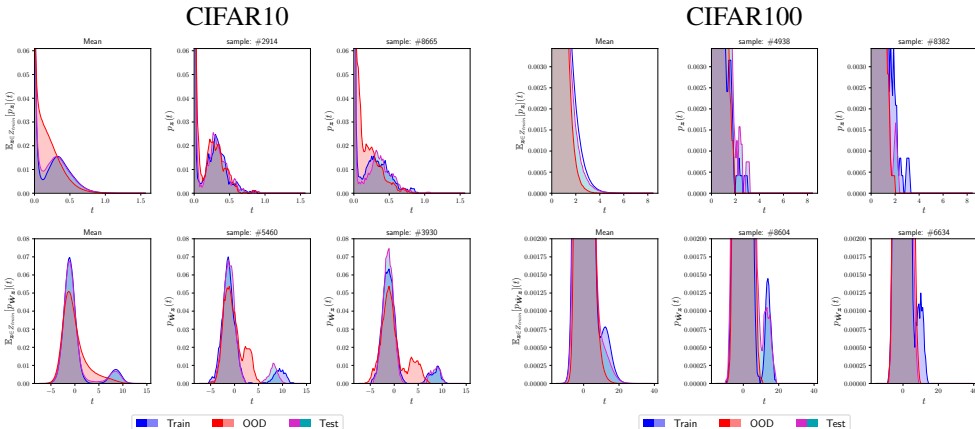

Figure 7: Density plots for CIFAR10 and CIFAR100 of a ResNet18 trained for 300 epochs. We present the densities as in Figure 3, but this time we show it for more datasets and for two different samples $z_1, z_2$ in the penultimate and the perturbed logit space, respectively. The OOD set for the ID set CIFAR10 is CIFAR100 and vice versa for CIFAR100. The range of the $y$-axis is adjusted such that the differences between ID and OOD become visible. We therefore report the mean over the maximum density $\max_p$ of the penultimate dimensions to show, up to which value the maximum would go. We apply smoothing over $k_s$ neighbors in each plot and construct the histograms with $n_{\text{bins}}$ bins. We report the parameters in Table 9. The class clusters and the activation clusters are clearly visible for CIFAR10 and merge into the bigger cluster for CIFAR100, probably because of the lower class to non-class ratio. It is harder to see for CIFAR100, but for both datasets, it seems harder for the OOD data to sample in the class cluster or the activated feature cluster.

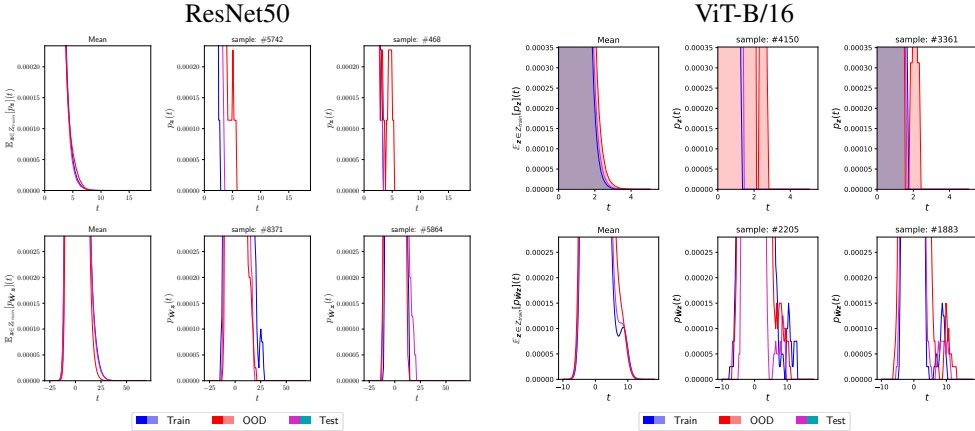

Figure 8: Density plots for ImageNet (ResNet50 and ViT-B/16). We chose SSB-hard as OOD set and apply the same plotting procedure as defined for CIFAR10/CIFAR100. For the respective plotting parameters, see Table 9. For ViT-B/16, the class clusters are distinguishable from the non-class clusters but not for ResNet50. Still, the difference between ID and OOD is captured in the higher activations which could explain why activation shaping Djurisic et al. (2022); Sun et al. (2021) works well for ImageNet.

Table 9: Plotting parameters: $s$ is the kernel size for the uniform kernel that was used for smoothing. and $\max_p = \max_t \mathbb{E}_{\boldsymbol{z} \in Z_{\text{train}}}[p_{\boldsymbol{z}}](t)$ denotes the maximum of the mean density of the penultimate densities $p_{\boldsymbol{t}}$. The perturbed densities $p_{\tilde{\boldsymbol{W}}\boldsymbol{z}}$ are scaled similarly.

|  | $n_{\text{bins}}$ | $s$ | $\max_p$ |
|---|---|---|---|
| CIFAR10 | 1000 | 2 | 364.22 |
| CIFAR100 | 1000 | 2 | 32.85 |
| ImageNet (ResNet50) | 100 | 4 | 1.78 |
| ImageNet (ViT-B/16) | 100 | 4 | 0.89 |

## A.5 Full CIFAR results

Table 10: Full CIFAR10 postprocessor results on the three ResNet18 checkpoints provided by OpenOOD trained with Cross Entropy and standard preprocessing. The $\pm$ indicates the standard deviation of all methods over three different model checkpoints.

| Method | CIFAR100 | | TIN | | $\mathcal{D}_{\text{near}}$ | |
|---|---|---|---|---|---|---|
| | AUROC ↑ | FPR95 ↓ | AUROC ↑ | FPR95 ↓ | AUROC ↑ | FPR95 ↓ |
| *Benchmark: CIFAR10 / Backbone: ResNet18* | | | | | | |
| OpenMax | 86.91±0.31 | 48.06±3.25 | 88.32±0.28 | 39.18±1.44 | 87.62±0.29 | 43.62±2.27 |
| MSP | 87.19±0.33 | 53.08±4.86 | 88.87±0.19 | 43.27±3.00 | 88.03±0.25 | 48.17±3.92 |
| TempScale | 87.17±0.40 | 55.81±5.07 | 89.00±0.23 | 46.11±3.63 | 88.09±0.31 | 50.96±4.32 |
| ODIN | 82.18±1.87 | 77.00±5.74 | 83.55±1.84 | 75.38±6.42 | 82.87±1.85 | 76.19±6.08 |
| MDS | 83.59±2.27 | 52.81±3.62 | 84.81±2.53 | 46.99±4.36 | 84.20±2.40 | 49.90±3.98 |
| MDSEns | 61.29±0.23 | 91.87±0.10 | 59.57±0.53 | 92.66±0.42 | 60.43±0.26 | 92.26±0.20 |
| RMDS | 88.83±0.35 | 43.86±3.49 | 90.76±0.27 | 33.91±1.39 | 89.80±0.28 | 38.89±2.39 |
| Gram | 58.33±4.49 | 91.68±2.24 | 58.98±5.19 | 90.06±1.59 | 58.66±4.83 | 90.87±1.91 |
| EBO | 86.36±0.58 | 66.60±4.46 | 88.80±0.36 | 56.08±4.83 | 87.58±0.46 | 61.34±4.63 |
| OpenGAN | 52.81±7.69 | 94.84±3.83 | 54.62±7.68 | 94.11±4.21 | 53.71±7.68 | 94.48±4.01 |
| GradNorm | 54.43±1.59 | 94.54±1.11 | 55.37±0.41 | 94.89±0.60 | 54.90±0.98 | 94.72±0.82 |
| ReAct | 85.93±0.83 | 67.40±7.34 | 88.29±0.44 | 59.71±7.31 | 87.11±0.61 | 63.56±7.33 |
| MLS | 86.31±0.59 | 66.59±4.44 | 88.72±0.36 | 56.06±4.82 | 87.52±0.47 | 61.32±4.62 |
| KLM | 77.89±0.75 | 90.55±5.83 | 80.49±0.85 | 85.18±7.60 | 79.19±0.80 | 87.86±6.37 |
| VIM | 87.75±0.28 | 49.19±3.15 | 89.62±0.33 | 40.49±1.55 | 88.68±0.28 | 44.84±2.31 |
| KNN | **89.73**±0.14 | 37.64±0.31 | **91.56**±0.26 | **30.37**±0.65 | **90.64**±0.20 | **34.01**±0.38 |
| DICE | 77.01±0.88 | 73.71±7.67 | 79.67±0.87 | 66.37±7.68 | 78.34±0.79 | 70.04±7.64 |
| RankFeat | 77.98±2.24 | 65.32±3.48 | 80.94±2.80 | 56.44±5.76 | 79.46±2.52 | 60.88±4.60 |
| ASH | 74.11±1.55 | 87.31±2.06 | 76.44±0.61 | 86.25±1.58 | 75.27±1.04 | 86.78±1.82 |
| SHE | 80.31±0.69 | 81.00±3.42 | 82.76±0.43 | 78.30±3.52 | 81.54±0.51 | 79.65±3.47 |
| GEN | 87.21±0.36 | 58.75±3.97 | 89.20±0.25 | 48.59±2.34 | 88.20±0.30 | 53.67±3.14 |
| **WeiPer+MSP** | 88.17±0.20 | 44.99±2.15 | 89.82±0.22 | 36.42±1.47 | 89.00±0.20 | 40.71±1.72 |
| **WeiPer+ReAct** | 88.02±0.47 | 47.87±5.09 | 89.63±0.37 | 37.81±5.30 | 88.83±0.41 | 42.84±5.11 |
| **WeiPer+KLD** | 89.70±0.27 | **37.42**±0.91 | 91.38±0.35 | 30.70±0.43 | 90.54±0.29 | 34.06±0.49 |

| Method | MNIST | | SVHN | | Textures | | Places365 | | $\mathcal{D}_{\text{far}}$ | |
|---|---|---|---|---|---|---|---|---|---|---|
| | AUROC ↑ | FPR95 ↓ | AUROC ↑ | FPR95 ↓ | AUROC ↑ | FPR95 ↓ | AUROC ↑ | FPR95 ↓ | AUROC ↑ | FPR95 ↓ |
| *Benchmark: CIFAR10 / Backbone: ResNet18* | | | | | | | | | | |
| NAC | 94.86±1.36 | 15.14±2.60 | **96.05**±0.47 | **14.33**±1.24 | **95.64**±0.44 | **17.03**±0.59 | **91.85**±0.28 | **26.73**±0.80 | **94.60**±0.50 | **18.31**±0.92 |
| OpenMax | 90.50±0.44 | 23.33±4.67 | 89.77±0.45 | 25.40±1.47 | 89.58±0.60 | 31.50±4.05 | 88.63±0.28 | 38.52±2.27 | 89.62±0.19 | 29.69±1.21 |
| MSP | 92.63±1.57 | 23.64±5.81 | 91.46±0.40 | 25.82±1.64 | 89.89±0.71 | 34.96±4.64 | 88.92±0.47 | 42.47±3.81 | 90.73±0.43 | 31.72±1.84 |
| TempScale | 93.11±1.77 | 23.53±7.05 | 91.66±0.52 | 26.97±2.65 | 90.01±0.74 | 38.16±5.89 | 89.11±0.52 | 45.27±4.50 | 90.97±0.52 | 33.48±2.39 |
| ODIN | 95.24±1.96 | 23.83±12.34 | 84.58±0.77 | 68.61±0.52 | 86.94±2.26 | 67.70±11.06 | 85.07±1.24 | 70.36±6.96 | 87.96±0.61 | 57.62±4.24 |
| MDS | 90.10±2.41 | 27.30±3.55 | 91.18±0.47 | 25.96±2.52 | 92.69±1.06 | 27.94±4.20 | 84.90±2.54 | 47.67±4.54 | 89.72±1.36 | 32.22±3.40 |
| MDSEns | **99.17**±0.41 | **1.30**±0.51 | 66.56±0.58 | 74.34±1.04 | 77.40±0.28 | 76.07±0.17 | 52.47±0.15 | 94.16±0.33 | 73.90±0.27 | 61.47±0.48 |
| RMDS | 93.22±0.80 | 21.49±2.32 | 91.84±0.26 | 23.46±1.48 | 92.23±0.23 | 25.25±0.53 | 91.51±0.11 | 31.20±0.28 | 92.20±0.21 | 25.35±0.73 |
| Gram | 72.64±2.34 | 70.30±8.96 | 91.52±4.45 | 33.91±17.35 | 62.34±8.27 | 94.64±2.71 | 60.44±3.41 | 90.49±1.93 | 71.73±3.20 | 72.34±6.73 |
| EBO | 94.32±2.53 | 24.99±12.93 | 91.79±0.98 | 35.12±6.11 | 89.47±0.70 | 51.82±6.11 | 89.25±0.78 | 54.85±6.52 | 91.21±0.92 | 41.69±5.32 |
| OpenGAN | 56.14±24.08 | 79.54±19.71 | 52.81±27.60 | 75.27±26.93 | 56.14±18.26 | 83.95±14.89 | 53.34±5.79 | 95.32±4.45 | 54.61±15.51 | 83.52±11.63 |
| GradNorm | 63.72±7.37 | 85.41±4.85 | 53.91±6.36 | 91.65±2.42 | 52.07±4.09 | 98.09±0.49 | 60.50±5.33 | 92.46±2.28 | 57.55±3.22 | 91.90±2.23 |
| ReAct | 92.81±3.03 | 33.77±18.00 | 89.12±3.19 | 50.23±15.98 | 89.38±1.49 | 51.42±11.42 | 90.35±0.78 | 44.20±3.35 | 90.42±1.41 | 44.90±8.37 |
| MLS | 94.15±2.48 | 25.06±12.87 | 91.69±0.94 | 35.09±6.09 | 89.41±0.71 | 51.73±6.13 | 89.14±0.76 | 54.84±6.51 | 91.10±0.89 | 41.68±5.27 |
| KLM | 85.00±2.04 | 76.22±12.09 | 84.99±1.18 | 59.47±7.06 | 82.35±0.33 | 81.95±9.95 | 78.37±0.33 | 95.58±2.12 | 82.68±0.21 | 78.31±4.84 |
| VIM | 94.76±0.38 | 18.36±1.42 | 94.50±0.48 | 19.29±0.41 | 95.15±0.34 | 21.14±1.83 | 89.49±0.39 | 41.43±2.17 | 93.48±0.24 | 25.05±0.52 |
| KNN | 94.26±0.38 | 20.05±1.36 | 92.67±0.30 | 22.60±1.26 | 93.16±0.24 | 24.06±0.55 | 91.77±0.23 | 30.38±0.63 | 92.96±0.14 | 24.27±0.40 |
| DICE | 90.37±5.97 | 30.83±10.54 | 90.02±1.77 | 36.61±4.74 | 81.86±2.35 | 62.42±4.79 | 74.67±4.98 | 77.19±12.60 | 84.23±1.89 | 51.76±4.42 |
| RankFeat | 75.87±5.22 | 61.86±12.78 | 68.15±7.44 | 64.49±7.38 | 73.46±6.49 | 59.71±9.79 | 85.99±3.04 | 43.70±7.39 | 75.87±5.06 | 57.44±7.99 |
| ASH | 83.16±4.66 | 70.00±10.56 | 73.46±6.41 | 83.64±6.48 | 77.45±2.39 | 84.59±1.74 | 79.89±3.69 | 77.89±7.28 | 78.49±2.58 | 79.03±4.22 |
| SHE | 90.43±4.76 | 42.22±20.59 | 86.38±1.32 | 62.74±4.01 | 81.57±1.21 | 84.60±5.30 | 82.89±1.22 | 76.36±5.32 | 85.32±1.43 | 66.48±5.98 |
| GEN | 93.83±2.14 | 23.00±7.75 | 91.97±0.66 | 28.14±2.59 | 90.14±0.76 | 40.74±6.61 | 89.46±0.65 | 47.03±3.22 | 91.35±0.69 | 34.73±1.58 |
| **WeiPer+MSP** | 92.76±1.49 | 24.21±4.35 | 92.05±0.60 | 24.85±1.34 | 91.29±0.58 | 28.35±2.80 | 89.57±0.39 | 38.06±2.96 | 91.42±0.44 | 28.87±1.29 |
| **WeiPer+ReAct** | 92.42±1.58 | 25.33±5.17 | 91.42±1.33 | 28.63±6.44 | 91.18±0.87 | 28.38±6.45 | 89.92±0.47 | 35.64±2.46 | 91.23±0.62 | 29.50±3.35 |
| **WeiPer+KLD** | 94.40±1.47 | 19.98±4.08 | 94.30±0.41 | 19.48±4.08 | 93.20±0.46 | 19.48±0.18 | 90.60±0.24 | 31.88±1.20 | 93.12±0.34 | 23.72±0.79 |

Table 11: Full CIFAR100 postprocessor results on the three ResNet18 checkpoints provided by OpenOOD trained with Cross Entropy and standard preprocessing.

| Method | CIFAR10 | | TIN | | $\mathcal{D}_{\textbf{near}}$ | |
|---|---|---|---|---|---|---|
| | AUROC ↑ | FPR95 ↓ | AUROC ↑ | FPR95 ↓ | AUROC ↑ | FPR95 ↓ |
| *Benchmark: CIFAR100 / Backbone: ResNet18* | | | | | | |
| OpenMax | 74.38±0.37 | 60.17±0.97 | 78.44±0.14 | 52.99±0.51 | 76.41±0.25 | 56.58±0.73 |
| MSP | 78.47±0.07 | 58.91±0.93 | 82.07±0.17 | 50.70±0.34 | 80.27±0.11 | 54.80±0.33 |
| TempScale | 79.02±0.06 | **58.72**±0.81 | 82.79±0.09 | 50.26±0.16 | 80.90±0.07 | 54.49±0.48 |
| ODIN | 78.18±0.14 | 60.64±0.56 | 81.63±0.08 | 55.19±0.57 | 79.90±0.11 | 57.91±0.51 |
| MDS | 55.87±0.22 | 88.00±0.49 | 61.50±0.28 | 79.05±1.22 | 58.69±0.09 | 83.53±0.60 |
| MDSEns | 43.85±0.31 | 95.94±0.16 | 48.78±0.19 | 95.82±0.12 | 46.31±0.24 | 95.88±0.04 |
| RMDS | 77.75±0.19 | 61.37±0.24 | 82.55±0.02 | 49.56±0.90 | 80.15±0.11 | 55.46±0.41 |
| Gram | 49.41±0.58 | 92.71±0.64 | 53.91±1.58 | 91.85±0.86 | 51.66±0.77 | 92.28±0.29 |
| EBO | 79.05±0.11 | 59.21±0.75 | 82.76±0.08 | 52.03±0.50 | 80.91±0.08 | 55.62±0.61 |
| OpenGAN | 63.23±2.44 | 78.83±3.94 | 68.74±2.29 | 74.21±1.25 | 65.98±1.26 | 76.52±2.59 |
| GradNorm | 70.32±0.20 | 84.30±0.36 | 69.95±0.79 | 86.85±0.62 | 70.13±0.47 | 85.58±0.46 |
| ReAct | 78.65±0.05 | 61.30±0.43 | 82.88±0.08 | 51.47±0.47 | 80.77±0.05 | 56.39±0.34 |
| MLS | 79.21±0.10 | 59.11±0.64 | 82.90±0.05 | 51.83±0.70 | 81.05±0.07 | 55.47±0.66 |
| KLM | 73.91±0.25 | 84.77±2.95 | 79.22±0.28 | 71.07±0.59 | 76.56±0.25 | 77.92±1.31 |
| VIM | 72.21±0.41 | 70.59±0.43 | 77.76±0.16 | 54.66±0.42 | 74.98±0.13 | 62.63±0.27 |
| KNN | 77.02±0.25 | 72.80±0.44 | 83.34±0.16 | 49.65±0.37 | 80.18±0.15 | 61.22±0.14 |
| DICE | 78.04±0.32 | 60.98±1.10 | 80.72±0.30 | 54.93±0.53 | 79.38±0.23 | 57.95±0.53 |
| RankFeat | 58.04±2.36 | 82.78±1.56 | 65.72±0.22 | 78.40±0.95 | 61.88±1.28 | 80.59±1.10 |
| ASH | 76.48±0.30 | 68.06±0.44 | 79.92±0.20 | 63.35±0.90 | 78.20±0.15 | 65.71±0.24 |
| SHE | 78.15±0.03 | 60.41±0.51 | 79.74±0.36 | 57.74±0.73 | 78.95±0.18 | 59.07±0.25 |
| GEN | **79.38**±0.04 | 58.87±0.69 | 83.25±0.13 | 49.98±0.05 | 81.31±0.08 | 54.42±0.33 |
| **WeiPer+MSP** | 79.24±0.20 | 59.69±1.20 | 83.39±0.06 | 49.28±0.26 | 81.32±0.10 | 54.49±0.63 |
| **WeiPer+ReAct** | 79.00±0.18 | 60.41±1.10 | 83.40±0.05 | 49.65±0.33 | 81.20±0.09 | 55.03±0.52 |
| **WeiPer+KLD** | 79.20±0.10 | 59.90±0.62 | **83.54**±0.07 | **48.78**±0.24 | **81.37**±0.02 | 54.34±0.29 |

| Method | MNIST | | SVHN | | Textures | | Places365 | | $\mathcal{D}_{\textbf{far}}$ | |
|---|---|---|---|---|---|---|---|---|---|---|
| | AUROC ↑ | FPR95 ↓ | AUROC ↑ | FPR95 ↓ | AUROC ↑ | FPR95 ↓ | AUROC ↑ | FPR95 ↓ | AUROC ↑ | FPR95 ↓ |
| *Benchmark: CIFAR100 / Backbone: ResNet18* | | | | | | | | | | |
| NAC | 93.15±1.63 | 21.97±6.62 | 92.40±1.26 | 24.39±4.66 | **89.32**±0.55 | **40.65**±1.94 | 73.05±0.68 | 73.57±1.16 | **86.98**±0.37 | **40.14**±1.86 |
| OpenMax | 76.01±1.39 | 53.82±4.74 | 82.07±1.53 | 53.20±1.78 | 80.56±0.09 | 56.12±1.91 | 79.29±0.40 | 54.85±1.42 | 79.48±0.41 | 54.50±0.68 |
| MSP | 76.08±1.86 | 57.23±4.68 | 78.42±0.89 | 59.07±2.53 | 77.32±0.71 | 61.88±1.28 | 79.22±0.29 | 56.62±0.87 | 77.76±0.44 | 58.70±1.06 |
| TempScale | 77.27±1.85 | 56.05±4.61 | 79.79±1.05 | 57.71±2.68 | 78.11±0.72 | 61.56±1.43 | 79.80±0.25 | 56.46±0.94 | 78.74±0.51 | 57.94±1.14 |
| ODIN | 83.79±1.31 | 45.94±3.29 | 74.54±0.76 | 67.41±3.88 | 79.33±1.08 | 62.37±2.96 | 79.45±0.26 | 59.71±0.92 | 79.28±0.21 | 58.86±0.79 |
| MDS | 67.47±0.81 | 71.72±2.94 | 70.68±6.40 | 67.21±6.09 | 76.26±0.69 | 70.49±2.48 | 63.15±0.49 | 79.61±0.34 | 69.39±1.39 | 72.26±1.56 |
| MDSEns | **98.21**±0.78 | **2.83**±0.86 | 53.76±1.63 | 82.57±2.58 | 69.75±1.14 | 84.94±0.83 | 42.27±0.73 | 96.61±0.17 | 66.00±0.69 | 66.74±1.04 |
| RMDS | 79.74±2.49 | 52.05±6.28 | 84.89±1.10 | 51.65±3.68 | 83.65±0.51 | 53.99±1.06 | **83.40**±0.46 | **53.57**±0.43 | 82.92±0.42 | 52.81±0.63 |
| Gram | 80.71±4.15 | 53.53±7.45 | **95.55**±0.60 | **20.06**±1.96 | 70.79±1.32 | 89.51±2.54 | 46.38±1.21 | 94.67±0.60 | 73.36±1.08 | 64.44±2.37 |
| EBO | 79.18±1.37 | 52.62±3.83 | 82.03±1.74 | 53.62±3.14 | 78.35±0.83 | 62.35±2.06 | 79.52±0.23 | 57.75±0.86 | 79.77±0.61 | 56.59±1.38 |
| OpenGAN | 68.14±18.78 | 63.09±23.25 | 68.40±2.15 | 70.35±2.06 | 65.84±3.43 | 74.77±1.78 | 69.13±7.08 | 73.75±8.32 | 67.88±7.16 | 70.49±7.38 |
| GradNorm | 65.35±1.12 | 86.97±1.44 | 76.95±4.73 | 69.90±7.94 | 64.58±0.13 | 92.51±0.61 | 69.69±0.17 | 85.32±0.44 | 69.14±1.05 | 83.68±1.92 |
| ReAct | 78.37±1.59 | 56.04±5.66 | 83.01±0.97 | 50.41±2.02 | 80.15±0.46 | 55.04±0.82 | 80.03±0.11 | 55.30±0.41 | 80.39±0.49 | 54.20±1.56 |
| MLS | 78.91±1.47 | 52.95±3.82 | 81.65±1.49 | 53.90±3.04 | 78.39±0.84 | 62.39±2.13 | 79.75±0.24 | 57.68±0.91 | 79.67±0.57 | 56.73±1.33 |
| KLM | 74.15±2.59 | 73.09±6.67 | 79.34±0.44 | 50.30±7.04 | 75.77±0.45 | 81.80±5.80 | 75.70±0.24 | 81.40±1.58 | 76.24±0.52 | 71.65±2.01 |
| VIM | 81.89±1.02 | 48.32±1.07 | 83.14±3.71 | 46.22±5.46 | 85.91±0.78 | 46.86±2.29 | 75.85±0.37 | 61.57±0.77 | 81.70±0.62 | 50.74±1.00 |
| KNN | 82.36±1.52 | 48.58±4.67 | 84.15±1.09 | 51.75±3.12 | 83.66±0.83 | 53.56±2.32 | 79.43±0.47 | 60.70±1.03 | 82.40±0.17 | 53.65±0.28 |
| DICE | 79.86±1.89 | 51.79±3.67 | 84.22±2.00 | 49.58±3.32 | 77.63±0.34 | 64.23±1.61 | 78.33±0.66 | 59.39±1.25 | 80.01±0.18 | 56.25±0.60 |
| RankFeat | 63.03±3.86 | 75.01±5.83 | 72.14±1.39 | 58.49±2.30 | 69.40±3.08 | 66.87±3.80 | 63.82±1.83 | 77.42±1.96 | 67.10±1.42 | 69.45±1.01 |
| ASH | 77.23±0.46 | 66.58±3.88 | 85.60±1.40 | 46.00±2.67 | 80.72±0.70 | 61.27±2.74 | 78.76±0.16 | 62.95±0.99 | 80.58±0.66 | 59.20±2.46 |
| SHE | 76.76±1.07 | 58.78±2.70 | 80.97±3.98 | 59.15±7.61 | 73.64±1.28 | 73.29±3.22 | 76.30±0.51 | 65.24±0.98 | 76.92±1.16 | 64.12±2.70 |
| GEN | 78.29±2.05 | 53.92±5.71 | 81.41±1.50 | 55.45±2.76 | 78.74±0.81 | 61.23±1.40 | 80.28±0.27 | 56.25±1.01 | 79.68±0.75 | 56.71±1.59 |
| **WeiPer+MSP** | 79.81±1.37 | 52.31±3.65 | 80.90±1.22 | 59.31±1.96 | 78.87±0.62 | 59.56±1.85 | 80.22±0.17 | 56.82±0.50 | 79.95±0.66 | 57.00±1.40 |
| **WeiPer+ReAct** | 79.09±1.36 | 53.91±3.98 | 81.90±0.64 | 56.00±3.52 | 79.77±0.36 | 56.78±0.91 | 80.49±0.15 | 55.74±0.43 | 80.31±0.39 | 55.61±0.79 |
| **WeiPer+KLD** | 77.93±2.09 | 55.51±5.58 | 79.55±0.97 | 59.80±2.70 | 78.56±0.79 | 59.63±1.55 | 80.00±0.24 | 56.90±0.85 | 79.01±0.54 | 57.96±0.98 |

## A.6 Full ImageNet results

Table 12: Full ImageNet postprocessor results on ResNet50 trained with Cross Entropy and standard preprocessing. We achieve three out of five best AUROC performances outperforming the competition.

| Method | SSB-hard | | NINCO | | $\mathcal{D}_{\text{near}}$ | |
|---|---|---|---|---|---|---|
| | AUROC ↑ | FPR95 ↓ | AUROC ↑ | FPR95 ↓ | AUROC ↑ | FPR95 ↓ |
| *Benchmark: ImageNet-1K / Backbone: ResNet50* | | | | | | |
| OpenMax | 71.37 | 77.33 | 78.17 | 60.81 | 74.77 | 69.07 |
| MSP | 72.09 | 74.49 | 79.95 | 56.88 | 76.02 | 65.68 |
| TempScale | 72.87 | 73.90 | 81.41 | 55.10 | 77.14 | 64.50 |
| ODIN | 71.74 | 76.83 | 77.77 | 68.16 | 74.75 | 72.50 |
| MDS | 48.50 | 92.10 | 62.38 | 78.80 | 55.44 | 85.45 |
| MDSEns | 43.92 | 95.19 | 55.41 | 91.86 | 49.67 | 93.52 |
| RMDS | 71.77 | 77.88 | 82.22 | 52.20 | 76.99 | 65.04 |
| Gram | 57.39 | 89.39 | 66.01 | 83.87 | 61.70 | 86.63 |
| EBO | 72.08 | 76.54 | 79.70 | 60.58 | 75.89 | 68.56 |
| GradNorm | 71.90 | 78.24 | 74.02 | 79.54 | 72.96 | 78.89 |
| ReAct | 73.03 | 77.55 | 81.73 | 55.82 | 77.38 | 66.69 |
| MLS | 72.51 | 76.20 | 80.41 | 59.44 | 76.46 | 67.82 |
| KLM | 71.38 | 84.71 | 81.90 | 60.36 | 76.64 | 72.54 |
| VIM | 65.54 | 80.41 | 78.63 | 62.29 | 72.08 | 71.35 |
| KNN | 62.57 | 83.36 | 79.64 | 58.39 | 71.10 | 70.87 |
| DICE | 70.13 | 77.96 | 76.01 | 66.90 | 73.07 | 72.43 |
| RankFeat | 55.89 | 89.63 | 46.08 | 94.03 | 50.99 | 91.83 |
| ASH | 72.89 | 73.66 | 83.45 | 52.97 | 78.17 | 63.32 |
| SHE | 71.08 | 76.30 | 76.49 | 69.72 | 73.78 | 73.01 |
| GEN | 72.01 | 75.73 | 81.70 | 54.90 | 76.85 | 65.32 |
| **WeiPer+MSP** | 73.01 | 75.16 | 82.35 | 52.53 | 77.68 | 63.84 |
| **WeiPer+ReAct** | 71.20 | 80.39 | 82.49 | 53.36 | 76.85 | 66.87 |
| **WeiPer+KLD** | **74.73** | 74.12 | **85.37** | **48.67** | **80.05** | **61.39** |

| Method | iNaturalist | | Textures | | Openimage-O | | $\mathcal{D}_{\text{far}}$ | |
|---|---|---|---|---|---|---|---|---|
| | AUROC ↑ | FPR95 ↓ | AUROC ↑ | FPR95 ↓ | AUROC ↑ | FPR95 ↓ | AUROC ↑ | FPR95 ↓ |
| *Benchmark: ImageNet-1K / Backbone: ResNet50* | | | | | | | | |
| NAC | 96.52 | - | 97.90 | - | 91.45 | - | 95.29 | - |
| OpenMax | 92.05 | 25.29 | 88.10 | 40.26 | 87.62 | 37.39 | 89.26 | 34.31 |
| MSP | 88.41 | 43.34 | 82.43 | 60.87 | 84.86 | 50.13 | 85.23 | 51.45 |
| TempScale | 90.50 | 37.63 | 84.95 | 56.90 | 87.22 | 45.40 | 87.56 | 46.64 |
| ODIN | 91.17 | 35.98 | 89.00 | 49.24 | 88.23 | 46.67 | 89.47 | 43.96 |
| MDS | 63.67 | 73.81 | 89.80 | 42.79 | 69.27 | 72.15 | 74.25 | 62.92 |
| MDSEns | 61.82 | 84.23 | 79.94 | 73.31 | 60.80 | 90.77 | 67.52 | 82.77 |
| RMDS | 87.24 | 33.67 | 86.08 | 48.80 | 85.84 | 40.27 | 86.38 | 40.91 |
| Gram | 76.67 | 67.89 | 88.02 | 58.80 | 74.43 | 75.39 | 79.71 | 67.36 |
| EBO | 90.63 | 31.30 | 88.70 | 45.77 | 89.06 | 38.09 | 89.47 | 38.39 |
| GradNorm | 93.89 | 32.03 | 92.05 | 43.27 | 84.82 | 68.46 | 90.25 | 47.92 |
| ReAct | 96.34 | 16.72 | 92.79 | 29.64 | 91.87 | 32.58 | 93.67 | 26.31 |
| MLS | 91.17 | 30.61 | 88.39 | 46.17 | 89.17 | 37.88 | 89.57 | 38.22 |
| KLM | 90.78 | 38.52 | 84.72 | 52.40 | 87.30 | 48.89 | 87.60 | 46.60 |
| VIM | 89.56 | 30.68 | **97.97** | **10.51** | 90.50 | 32.82 | 92.68 | 24.67 |
| KNN | 86.41 | 40.80 | 97.09 | 17.31 | 87.04 | 44.27 | 90.18 | 34.13 |
| DICE | 92.54 | 33.37 | 92.04 | 44.28 | 88.26 | 47.83 | 90.95 | 41.83 |
| RankFeat | 40.06 | 94.40 | 70.90 | 76.84 | 50.83 | 90.26 | 53.93 | 87.17 |
| ASH | 97.07 | 14.04 | 96.90 | 15.26 | **93.26** | **29.15** | **95.74** | **19.49** |
| SHE | 92.65 | 34.06 | 93.60 | 35.27 | 86.52 | 55.02 | 90.92 | 41.45 |
| GEN | 92.44 | 26.10 | 87.59 | 46.22 | 89.26 | 34.50 | 89.76 | 35.61 |
| **WeiPer+MSP** | 92.44 | 29.77 | 86.62 | 55.16 | 88.94 | 39.75 | 89.33 | 41.56 |
| **WeiPer+ReAct** | 95.75 | 21.03 | 91.88 | 34.95 | 91.64 | 33.53 | 93.09 | 29.83 |
| **WeiPer+KLD** | **97.49** | **13.59** | 96.18 | 22.17 | 92.94 | 30.49 | 95.54 | 22.08 |

Table 13: Full ImageNet postprocessor results on ViT/16-B trained with Cross Entropy and standard preprocessing.

| Method | SSB-hard | | NINCO | | $\mathcal{D}_{\text{near}}$ | |
|---|---|---|---|---|---|---|
| | AUROC ↑ | FPR95 ↓ | AUROC ↑ | FPR95 ↓ | AUROC ↑ | FPR95 ↓ |
| *Benchmark: ImageNet-1K / Backbone: ViT16-B* | | | | | | |
| OpenMax | 68.60 | 89.19 | 78.68 | 88.33 | 73.64 | 88.76 |
| MSP | 68.94 | 86.41 | 78.11 | 77.28 | 73.52 | 81.85 |
| TempScale | 68.55 | 87.35 | 77.80 | 81.88 | 73.18 | 84.62 |
| MDS | 71.57 | 83.47 | 86.52 | 48.77 | 79.04 | 66.12 |
| RMDS | **72.87** | 84.52 | **87.31** | **46.20** | **80.09** | **65.36** |
| EBO | 58.80 | 92.24 | 66.02 | 94.14 | 62.41 | 93.19 |
| GradNorm | 42.96 | 93.62 | 35.60 | 95.81 | 39.28 | 94.71 |
| ReAct | 63.10 | 90.46 | 75.43 | 78.51 | 69.26 | 84.49 |
| MLS | 64.20 | 91.52 | 72.40 | 92.97 | 68.30 | 92.25 |
| KLM | 68.14 | 88.35 | 80.68 | 66.14 | 74.41 | 77.25 |
| VIM | 69.42 | 90.04 | 84.64 | 57.41 | 77.03 | 73.73 |
| KNN | 65.98 | 86.22 | 82.25 | 54.73 | 74.11 | 70.47 |
| DICE | 59.05 | 89.77 | 71.67 | 81.10 | 65.36 | 85.44 |
| ASH | 53.90 | 93.50 | 52.51 | 95.37 | 53.21 | 94.43 |
| SHE | 68.04 | 85.73 | 84.18 | 56.02 | 76.11 | 70.88 |
| GEN | 70.09 | **82.23** | 82.51 | 59.33 | 76.30 | 70.78 |
| **WeiPer+MSP** | 68.98 | 85.09 | 80.66 | 64.85 | 74.82 | 74.97 |
| **WeiPer+ReAct** | 68.52 | 85.48 | 81.07 | 62.67 | 74.79 | 74.08 |
| **WeiPer+KLD** | 68.26 | 85.60 | 81.73 | 60.45 | 75.00 | 73.02 |

| Method | iNaturalist | | Textures | | Openimage-O | | $\mathcal{D}_{\text{far}}$ | |
|---|---|---|---|---|---|---|---|---|
| | AUROC ↑ | FPR95 ↓ | AUROC ↑ | FPR95 ↓ | AUROC ↑ | FPR95 ↓ | AUROC ↑ | FPR95 ↓ |
| *Benchmark: ImageNet-1K / Backbone: ViT16-B* | | | | | | | | |
| NAC | 93.72 | - | **94.17** | - | 91.58 | - | **93.16** | - |
| OpenMax | 94.93 | 19.62 | 73.07 | 40.26 | 87.36 | 73.82 | 89.27 | 55.50 |
| MSP | 88.19 | 42.40 | 85.06 | 56.46 | 84.86 | 56.19 | 86.04 | 51.69 |
| TempScale | 88.54 | 43.09 | 85.39 | 58.16 | 85.04 | 59.98 | 86.32 | 53.74 |
| MDS | 96.01 | 20.64 | 89.41 | 38.91 | **92.38** | 30.35 | 92.60 | 29.97 |
| RMDS | **96.10** | **19.47** | 89.38 | 37.22 | 92.32 | **29.57** | 92.60 | 28.76 |
| EBO | 79.30 | 83.56 | 81.17 | 83.66 | 76.48 | 88.82 | 78.98 | 85.35 |
| GradNorm | 42.42 | 91.16 | 44.99 | 92.25 | 37.82 | 94.53 | 41.75 | 92.65 |
| ReAct | 86.11 | 48.25 | 86.66 | 55.88 | 84.29 | 57.67 | 85.69 | 53.93 |
| MLS | 85.29 | 72.94 | 83.74 | 78.94 | 81.60 | 85.82 | 83.54 | 79.23 |
| KLM | 89.59 | 43.48 | 86.49 | 50.12 | 87.03 | 51.75 | 87.70 | 48.45 |
| VIM | 95.72 | 17.59 | 90.61 | 40.35 | 92.18 | 29.61 | 92.84 | 29.18 |
| KNN | 91.46 | 27.75 | 91.12 | 33.23 | 89.86 | 34.82 | 90.81 | 31.93 |
| DICE | 82.50 | 47.90 | 82.21 | 54.83 | 82.22 | 52.57 | 82.31 | 51.77 |
| ASH | 50.62 | 97.02 | 48.53 | 98.50 | 55.51 | 94.79 | 51.56 | 96.77 |
| SHE | 93.57 | 22.16 | 92.65 | 25.63 | 91.04 | 33.57 | 92.42 | 27.12 |
| GEN | 93.54 | 22.92 | 90.23 | 38.30 | 90.27 | 35.47 | 91.35 | 32.23 |
| **WeiPer+MSP** | 91.23 | 35.55 | 88.08 | 48.62 | 88.15 | 46.30 | 89.15 | 43.49 |
| **WeiPer+ReAct** | 91.49 | 33.04 | 88.31 | 47.37 | 88.56 | 43.26 | 89.45 | 41.22 |
| **WeiPer+KLD** | 92.09 | 29.32 | 89.36 | 46.10 | 89.51 | 39.05 | 90.32 | 38.16 |

## A.7 Compute resources

All experiments are conducted on a local machine with the following key specifications: AMD EPYC 7543 (32-Core Processor) with 256GB RAM and 1x NVIDIA RTX A5000 (24GB VRAM). To streamline the experimental process, we pre-compute the penultimate output of each backbone model and dataset combination. This is possible as we do not alter the training objective for our evaluation, achieving a reduction of disk usage to <2.8GB when stored as FP16 tensors compared to sum of the original dataset sizes of CIFAR10, CIFAR100 and ImageNet-1K.

For all benchmarks, the 24GB VRAM suffices for both, the model inference and postprocessor optimization. Depending on the chosen batch size, this offers a runtime / VRAM trade-off and is therefore well achievable on smaller GPUs.

Excluding the inference step for the penultimate output, we report an inference time for the postprocessor optimization of 18 seconds for CIFAR10 and <10 minutes for ImageNet-1K per iteration. An iteration refers to processing the full dataset starting from the penultimate layer output. For the full duration without pre-processing, one would add the inference time of the respective ResNet[18/50] or ViT/16-B model.

