# OpenReview forum: "WeiPer: OOD Detection using Weight Perturbations of Class Projections"
_NeurIPS.cc/2024/Conference — NeurIPS 2024 poster_

### Official Review · Reviewer_ak1i · 2024-07-08

**Soundness:** 2
**Presentation:** 3
**Contribution:** 2
**Rating:** 6
**Confidence:** 3

**Summary:**

The paper introduces WeiPer, an post-hoc method for out-of-distribution (OOD) detection that leverages weight perturbations in the final layer of neural networks to enhance detection performance.

Contributions:

1. Introducing linear projections of the penultimate layer by perturbing the final layer's weights to improve OOD detection.

2. Discovering a fingerprint-like nature of in-distribution (ID) samples in both penultimate and newly perturbed spaces, leveraging this structure for a novel detection method.

3. Proposing a KL-divergence-based scoring function and evaluating it alongside MSP and ReAct methods, showing state-of-the-art performance on near OOD tasks using the OpenOOD benchmark.

**Strengths:**

Simple methods with good empirical results. Technical novelty is limited but this is not a concern if it outperforms previous methods.

**Weaknesses:**

**Related Work Missing Strong Baselines: Data Depths and Information Projections**

Data depths and information projections are gaining significant interest in the OOD detection community. However, these approaches are notably absent in the related work section. It is essential to address these gaps to provide a comprehensive overview of the field.

Relevant works include:

M. Darrin. "Unsupervised Layer-wise Score Aggregation for Textual OOD Detection."

M. Darrin. "Rainproof: An Umbrella To Shield Text Generators From Out-Of-Distribution Data."

M. Picot. "Adversarial Attack Detection Under Realistic Constraints."

M. Picot. "A Simple Unsupervised Data Depth-based Method to Detect Adversarial Images."

M. Picot. "A Halfspace-Mass Depth-Based Method for Adversarial Attack Detection." TMLR 2023.

P. Colombo. "Beyond Mahalanobis Distance for Textual OOD Detection." NeurIPS 2022.

P. Colombo. "Toward Stronger Textual Attack Detectors."

**Lack of Strong Baselines for Empirical Comparison**

The current work lacks strong baseline comparisons, particularly those involving data depth methods. This omission undermines the robustness of empirical evaluations. Incorporating these baselines is crucial for a fair and thorough comparison with existing methods.

Key references that should be considered for baseline comparisons include:

E. Gomes. "A Functional Perspective on Multi-Layer Out-of-Distribution Detection."
M. Picot. "A Simple Unsupervised Data Depth-based Method to Detect Adversarial Images."
M. Picot. "A Halfspace-Mass Depth-Based Method for Adversarial Attack Detection." TMLR 2023.

**Questions:**

1. Why did you not mention the data depths and only rely on Mahanalobis?
2. Why not use information projections?
3. Can you compare against the baselines previously introduced?

**I thank the authors for their rebuttal, i have increased my grade.**

---

> ### Author Rebuttal · Authors · 2024-08-07
>
> We thank the reviewer for reading our paper and analyzing the strengths and weaknesses of our approach.
>
> > "Related Work Missing Strong Baselines: Data Depths and Information Projections"
>
> Many  methods have been developed for OOD detection, even when only considering image classification. The papers the reviewer referenced mainly concern adversarial attack detection and textual OOD detection. For clarity and readability, we chose the scope of our related work section to include only image OOD detection, and we are aware that it comes short of many impactful publications in related fields. In terms of methodology, [1] and [2] are partially similar to our approach. They also use an f-divergence as a score function (although on the softmax output) and compare input distributions to train set distributions. We agree that these works should be mentioned and will add a paragraph to the Related Work section about methods that utilize f-divergences, including the papers [1] and [2].
>
> > Why did you not mention the data depths and only rely on Mahanalobis?
>
> Our method is not based on Mahalanobis distance. We cited publications working with Mahalanobis distance, but we are not aware of other methods that consider data depth. Our KLD method is a density-based score function. Although density and data depth are related concepts, we are unaware of a closer connection to data depth that would justify mentioning it.
>
> > Why not use information projections?
>
> We tried to formalize the score as information projection, but early testing showed that comparing to the mean training distribution was superior to comparing to the minimal distribution. Since conducting rigorous experiments on that matter did not fit our paper's scope, it is possible that different information measures, e.g., a different f-divergence and other ways of choosing a distribution to compare to, perform better than our choice. This is a promising idea for future work.
>
>
> > Can you compare against the baselines previously introduced?
>
> As we chose OpenOOD as a benchmark framework, we are only comparing methods that have already been evaluated there. Because of time and resource constraints, we were not able to compare it to other work originating from OOD detection on images or any other modality. The papers the reviewer mentioned mainly introduce adversarial attack detectors or textual OOD detectors, and we are unaware of their application to image OOD detection or that they are evaluated on OpenOOD. The only evaluation that overlaps with [3] is the ViT-B/16 on far OOD ImageNet benchmarks. We agree with OpenOOD [4] that the field of OOD detection is suffering from inconsistency in evaluation, as it is not possible to compare every method when results are reported on different datasets/models/checkpoints/training algorithms.
>
> [1] Darrin et al. "Rainproof: An Umbrella To Shield Text Generators From Out-Of-Distribution Data."
>
> [2] Picot et al. "Adversarial Attack Detection Under Realistic Constraints."
>
> [3] Gomes et al. "A Functional Perspective on Multi-Layer Out-of-Distribution Detection."
>
> [4] Zhang et al. “OpenOOD v1.5: Enhanced Benchmark for Out-of-Distribution Detection”

---

### Official Review · Reviewer_bFW6 · 2024-07-11

**Soundness:** 3
**Presentation:** 3
**Contribution:** 3
**Rating:** 7
**Confidence:** 5

**Summary:**

This work proposes a component, WeiPer, that can benefit OOD detection. WeiPer adds random perturbations (sampled from standard normal) to the class projection weight vectors and essentially expands the output dimension (compared to the original logit space). WeiPer can be combined with multiple existing OOD detection scoring functions (e.g., MSP, ReAct). The authors further propose a KL-divergence-based scoring mechanism that works particularly well with WeiPer. Experiments show that WeiPer+KLD yields state-of-the-art results on the challenging OpenOOD benchmarks, including the near-OOD one on ImageNet-1K.

**Strengths:**

1. The proposed WeiPer is to my knowledge novel. It also has generality since it can be combined with many existing OOD scoring functions.

2. Extensive analyses are presented along the introduction of the method, which well justifies each design choice and makes the underlying intuition/insight clear.

3. Most importantly, unlike many other papers that use arbitrary or easy OOD benchmarks for evaluation, this work demonstrates notable improvements on the challenging OpenOOD, especially with the ImageNet-1K near-OOD benchmark (with ~2% AUROC increase over the previous SOTA ASH).

**Weaknesses:**

A few weaknesses have been discussed by the authors in Sec. 4 and 5, e.g., WeiPer is less powerful on ViT and could induce higher memory consumption.

**Questions:**

Since WeiPer essentially expands the output logit space and is expected to encode richer information, is it possible to leverage WeiPer for other tasks such as OOD generalization, beyond OOD detection?

**Limitations:**

The authors adequately addressed the limitations.

---

> ### Author Rebuttal · Authors · 2024-08-07
>
> We thank the reviewer for taking the time to assess our paper and following up with a thought-provoking question.
>
> > Since WeiPer essentially expands the output logit space and is expected to encode richer information, is it possible to leverage WeiPer for other tasks such as OOD generalization, beyond OOD detection?
>
> This is an interesting direction for future research. As [1] showed, the penultimate features or more precisely the neural activation states with narrow coverage, i.e. peaky distributions, lead to weak OOD generalization abilities. The Neural Collapse theory [2] states that with continued training, the features converge to the class mean. We also observe that thereby the activation distributions become more narrow (see supplementary material `ResNet18_penultimate_layer_resize.gif`). This would result in the mean activation distribution and WeiPer space distribution becoming peaky. We believe that this could be connected to overfitting and measuring this could be effective in selecting more robust models. WeiPer could assist in detecting the collapse in the logit layer as the features converge to the class means not only along the weight directions.
>
> [1] Liu et al.,  Neuron activation coverage: Rethinking out-of-distribution detection and generalization. 2024
> [2] Papyan et al., Prevalence of neural collapse during the terminal phase of deep learning training. 2020

---

> > ### Comment · Reviewer_bFW6 · 2024-08-07
> > **Thanks for the rebuttal**
> >
> > I have read the rebuttal and have no further questions. I maintain my score.

---

### Official Review · Reviewer_T1ji · 2024-07-13

**Soundness:** 4
**Presentation:** 4
**Contribution:** 3
**Rating:** 7
**Confidence:** 2

**Summary:**

The manuscript proposes a post-hoc OOD detection method, which is broadly applicable and can improve existing OOD detection methods.

**Strengths:**

- The methodology is post-hoc, making it more practical.

- Results for near OOD evaluation are promising.

- The method can be combined with existing OOD detection strategies and improve their performance, thus increasing the potential impact of this work.

**Weaknesses:**

Since memory usage is a limitation of the proposed methodology, the manuscript should present a comparison of computational cost (time and memory) between the multiple OOD detection methodologies that are considered.

**Questions:**

How does your methodology compare to the state-of-the-art in terms of required memory and time?

**Limitations:**

The authors clearly stated the method's limitations.

---

> ### Author Rebuttal · Authors · 2024-08-07
>
> We thank reviewer for examining our paper, highlighting strengths, and indicating weaknesses to improve our contribution.
>
> We provide a memory and time analysis comparing WeiPer+KLD to its closest competitors in Table 1 and Table 2 in the rebuttal pdf. Our method is on par with the other methods. WeiPer+KLD currently uses the whole training set to calculate the mean distribution of the activations and the WeiPer space. We see that a far smaller sample is sufficient and will conduct an experiment in the camera-ready version of the paper.
>
> Unfortunately, we did not yet succeed in collecting all the results of the memory and time comparison on ImageNet, which are particularly insightful since WeiPer blows up the 1000 dimensional logit space even more than on CIFAR100 ($r\cdot 1000$ instead of $r\cdot 100$). We will provide them as soon as possible. Note that the memory information in Figure 4 might be misleading (e.g., 80.24 GiB for ImageNet with $r=100$) since these numbers would apply when we calculate all training, test, and OOD scores in parallel instead of using a batch-wise procedure. We will add this to the figure description.

---

> > ### Comment · Reviewer_T1ji · 2024-08-14
> >
> > I thank the authors for the new experiments. For the camera-ready, I suggest including comprehensive comparisons on time and memory consumption between the proposed method and diverse alternative ones.
> >
> > Moreover, I believe that the experiments using subsets of the training database, which the authors mentioned in their rebuttal, will be valuable for showing the practicality of their methodology.
> >
> > Overall, I am happy to increase my score to accept.

---

### Official Review · Reviewer_QiAi · 2024-07-13

**Soundness:** 3
**Presentation:** 2
**Contribution:** 3
**Rating:** 5
**Confidence:** 4

**Summary:**

The paper introduces "WeiPer," a method that improves existing out-of-distribution (OOD) detection techniques by perturbing class projection weights. This method leverages the class-discriminative ability of pre-trained neural network classifiers by introducing weight perturbations in the final fully connected layer, creating richer representations of the input. The authors demonstrate that this technique significantly enhances OOD detection performance across multiple benchmarks, especially in challenging near-OOD scenarios. Additionally, a KL-divergence-based scoring method is proposed to utilize the properties of the augmented WeiPer space, supported by theoretical motivations and empirical observations.

**Strengths:**

1. WeiPer introduces a novel and effective technique for OOD detection by incorporating weight perturbations, broadening the scope of current methods.

2. The paper provides extensive experimental evidence demonstrating the superior performance of the WeiPer method across multiple benchmarks and includes detailed ablation studies to validate the contributions of each component.

**Weaknesses:**

1. WeiPer introduces additional computational complexity and memory requirements, which might limit its applicability in resource-constrained environments.

2. The proposed method involves several hyperparameters that need tuning, potentially increasing the difficulty of practical implementation.

3. Compared with other methods, the superior performance is not very stable. While the method performs well across several benchmarks, additional validation on a wider variety of datasets would further demonstrate its generalizability.

**Questions:**

Can you provide more discussion on the motivation of the approach and its contribution to making the OOD field more robust?

**Limitations:**

The authors have addressed the limitations.

---

> ### Author Rebuttal · Authors · 2024-08-07
>
> We thank the reviewer for taking the time to assess our work.
>
> > WeiPer introduces additional computational complexity and memory requirements, which might limit its applicability in resource-constrained environments.
>
> That is true for virtually every method. WeiPer+KLD is comparable to other methods in both time and memory consumption (see Table 1 and 2 of the rebuttal PDF). Memory consumption and computational complexity scale linearly with $r$, all the other hyperparameters have no significant influence (see also the Limitations section in the paper).
> The proposed method involves several hyperparameters that need tuning, potentially increasing the difficulty of practical implementation.
> This is correct, however, most methods that perform competitively come with hyperparameters that need to be tuned. WeiPer+KLD has more hyperparameters than its competitors, but optimizing a single digit number of hyperparameters is common practice in Machine Learning. WeiPer+KLD can be optimized with OpenOODs hyperparameter search to a specific use case, and our experiments show that the optimization surface is smooth (see Figure 6 in the Appendix), i.e. other optimization methods should be easily applied.
>
> > Compared with other methods, the superior performance is not very stable. While the method performs well across several benchmarks, additional validation on a wider variety of datasets would further demonstrate its generalizability
>
> We disagree. In fact, WeiPer+KLD is the most robust method for near OOD detection tasks (harder than far) in our study (see Table 3 in the paper). We would like to mention that we use OpenOOD [2] who built a unified benchmark. OpenOOD includes 22 benchmarks (6 for CIFAR10, 6 for CIFAR100, 5 for ImageNet in ResNet50, and 5 for ImageNet on ViT-B/16), and each method is evaluated across all of them. Compared to other methods, we think WeiPer+KLD has been evaluated with the highest scrutiny. OOD detection is still hard, hence the seemingly mixed results.
>
> > Can you provide more discussion on the motivation of the approach and its contribution to making the OOD field more robust?
>
> The motivation of using the WeiPer space is that this extracts structural information that can be leveraged for ID / OOD detection. (see Theorem 1). The way most classifiers end up being trained is that  the OOD set reaches into the positive class cluster in a conical shape (see Figure 1). Since, with weight perturbations, we project the data from a different angle, we can exploit this property which should be fairly generic throughout datasets.
> We kindly ask the reviewer to clarify what they believe is missing in the motivation of our approach.
>
> [1] Zhang et al. “OpenOOD v1.5: Enhanced Benchmark for Out-of-Distribution Detection”
>
> [2] Sun et al. “Out-of-Distribution Detection with Deep Nearest Neighbors”, ICML 2022

---

> > ### Comment · Reviewer_QiAi · 2024-08-11
> >
> > The author has basically addressed my doubts. Regarding the more discussion about motivation I mentioned, I think a broader description of motivation and contribution can be given in the context of the task. I am happy to improve my score to Borderline Accept.

---

### Author Rebuttal · Authors · 2024-08-07

Memory and time comparison to other methods.

---

### Decision · Program_Chairs · 2024-09-25

**Decision:**

Accept (poster)

**Comment:**

The paper introduces a novel approach to enhance out-of-distribution detection by perturbing the weight vectors in class projections, expanding the output dimension, and enabling richer representation extraction from neural networks. The authors have presented a significant amount of empirical evidence demonstrating improved performance across challenging benchmarks, particularly in near-OOD scenarios.

The primary concerns raised in the reviews centered around the additional computational and memory requirements of the WeiPer method, its stability across various datasets, and the need for tuning multiple hyperparameters. The authors' rebuttals have addressed these concerns satisfactorily. Given the unanimous support, the decision is to accept this paper. The authors are encouraged to incorporate the suggested enhancements in their final version.